# Injectable human recombinant collagen matrices limit adverse remodeling and improve cardiac function after myocardial infarction

Sarah McLaughlin[1,2,8], Brian McNeill[1,8], James Podrebarac[1,2], Katsuhiro Hosoyama[1], Veronika Sedlakova [1], Gregory Cron[3], David Smyth[4], Richard Seymour[1], Keshav Goel [1], Wenbin Liang[2,5], Katey J. Rayner[6,7], Marc Ruel[1,2], Erik J. Suuronen[1,2,9]* & Emilio I. Alarcon [1,7,9]*

Despite the success of current therapies for acute myocardial infarction (MI), many patients still develop adverse cardiac remodeling and heart failure. With the growing prevalence of heart failure, a new therapy is needed that can prevent remodeling and support tissue repair. Herein, we report on injectable recombinant human collagen type I (rHCI) and type III (rHCIII) matrices for treating MI. Injecting rHCI or rHCIII matrices in mice during the late proliferative phase post-MI restores the myocardium's mechanical properties and reduces scar size, but only the rHCI matrix maintains remote wall thickness and prevents heart enlargement. rHCI treatment increases cardiomyocyte and capillary numbers in the border zone and the presence of pro-wound healing macrophages in the ischemic area, while reducing the overall recruitment of bone marrow monocytes. Our findings show functional recovery post-MI using rHCI by promoting a healing environment, cardiomyocyte survival, and less pathological remodeling of the myocardium.

[1] BioEngineering and Therapeutic Solutions (BEaTS), Division of Cardiac Surgery, University of Ottawa Heart Institute, 40 Ruskin street, Ottawa, ON K1Y4W7, Canada. [2] Department of Cellular & Molecular Medicine, University of Ottawa, 451 Smyth Road, Ottawa, ON K1H8M5, Canada. [3] Department of Radiology, Faculty of Medicine, University of Ottawa, 501 Smyth Road, Ottawa, ON K1H8L6, Canada. [4] Cardiac Function Laboratory, University of Ottawa Heart Institute, 40 Ruskin street, Ottawa, ON K1Y4W7, Canada. [5] Cardiac Electrophysiology Lab, University of Ottawa Heart Institute, 40 Ruskin street, Ottawa, ON K1Y4W7, Canada. [6] Cardiometabolic microRNA Laboratory, University of Ottawa Heart Institute, 40 Ruskin street, Ottawa, ON K1Y4W7, Canada. [7] Department of Biochemistry, Microbiology, and Immunology, University of Ottawa, 451 Smyth Road, Ottawa, ON K1H8M5, Canada. [8] These authors contributed equally: Sarah McLaughlin, Brian McNeill. [9] These authors jointly supervised: Erik J. Suuronen, Emilio I. Alarcon. *email: esuuronen@ottawaheart.ca; ealarcon@ottawaheart.ca

Cardiovascular diseases (CVD) are the number one cause of worldwide morbidity and are responsible for over 17 million deaths annually[1]. Coronary artery disease remains the most common CVD and is caused by atherosclerotic plaque accumulation in the epicardial arteries supplying the myocardium[2], which can restrict blood flow leading to myocardial infarction (MI)[1,3]. Surgical procedures that restore blood flow to the scarred cardiac muscle post-MI improve patient outcomes; however, they fail to restore cardiac function. On average, 10% of post-MI patients develop adverse ventricular remodeling that will ultimately lead to advanced heart failure[1], which has a 5-year mortality of ~50%[4]. Despite tremendous efforts in developing biological therapeutics to overcome cardiac muscle damage and dysfunction post-MI, it has become increasingly evident that small-molecule drugs, growth factors, and even cell-based therapies are currently suboptimal for restoring heart function[5–7]. The limited endogenous ability of the cardiac muscle to heal[8] and modification of the cardiac extracellular matrix (ECM)[9] are two key factors that inhibit repair and regeneration. Thus, a new therapy is needed that can simultaneously prevent adverse remodeling and provide a suitable ECM environment to support cell and tissue repair, as well as functional restoration.

Post-MI, ECM protein modification is initiated during the inflammatory phase, which differs from late-stage cardiac remodeling ECM proteins[10–12]. These changes disrupt cell–ECM interactions required for cell signaling, function, and survival[13,14]. Therefore, restoring the ECM within the infarcted myocardium may help limit adverse remodeling and ultimately improve cardiac function[15,16], and injectable biomaterials have been proposed as therapies for this purpose[17–21]. Although biomaterials can provide physical stability to the infarcted myocardium, this rather passive structural reinforcement alone may be insufficient to sustain cardiac function in the long-term. Therefore, an important feature of injectable biomaterials is that they can be designed to act as a biomimetic matrix for supporting cells and stimulating infarct repair[22–24]. Furthermore, the use of a biomimetic platform capable of promoting endogenous tissue repair is a critical feature for safe and effective clinical translation.

The healthy cardiac ECM is composed primarily of type I (70%) and type III (12%) collagens[25]. In animal MI models, biomaterials based on collagen have been able to provide mechanical support, improve angiogenesis and tissue integration, reduce inflammation and apoptosis, and limit negative remodeling and the loss of cardiac function[26–31]. Notably, the safety and efficacy of a porcine myocardial ECM hydrogel, composed primarily of collagens, was established in a pre-clinical pig MI model[19]. Despite the promise of biomaterials for cardiac therapy, there are still some limitations to consider. First, biomaterials tested so far have been of animal origin, which carry intrinsic batch-to-batch variability due to isolation protocols, and inherent immune risks (i.e., endotoxins)[32,33]. While biomaterials have prevented further loss of function when applied early post-MI[17–21], an injectable material that can increase cardiac function when delivered during the late proliferative phase post-MI is also needed.

Here, instead of using animal-derived components, we report the first injectable biomaterials made from recombinant human collagens type I (rHCI) and type III (rHCIII) in order to produce a clinically translatable material for cardiac therapy. Notably, these proteins have been previously used for the fabrication of corneal implants, which were demonstrated to be safe when transplanted in humans[34]. We investigate the pre-clinical performance of our injectable rHCI and rHCIII materials in a well-established mouse model of MI[35]. Intramyocardial injections of rHCI or rHCIII are administered to mouse hearts at 1-week post-MI (Supplementary Fig. 1 and Supplementary Video 1), which is a model for patients who have a delay in getting medical attention (e.g., remote location, or ignore early signs after heart attack) or who do not adequately respond to conventional treatment, and are at greater risk of ECM remodeling and developing heart failure[36,37]. This time-point constitutes the proliferative phase of infarction during which scar formation is initiated (mice: 2–7 days; humans: 4–14 days), and is a prime opportunity to intervene to limit pathological ECM remodeling and promote infarct repair[11,38–40]. The primary outcome is cardiac function determined by echocardiography 28 days after treatment, and we assess cardiac morphology, ventricular remodeling, vascularization, and inflammation in the MI heart in response to rHC treatment. In addition, we use in vitro studies to elucidate how macrophages interact with the rHC materials. Our results highlight the potential for a clinically relevant biomaterial based on human recombinant collagen to be used as therapy for the amelioration of cardiac function post-MI through the promotion of a wound healing environment, cardiomyocyte survival, and reduced pathological remodeling.

## Results

**Synthesis and characterization of rHC matrices.** The collagen formulations used in this work were developed as thermoresponsive matrices to ensure their retention within the myocardium upon intramyocardial injection. In designing the materials, the goal was to have a gelation time of less than 10 min to allow sufficient time for injection in liquid form with minimal time required for subsequent gelation. The same total concentration of collagens (rHCI and rHCIII) and the cross-linker agents N-ethyl-N-(3-dimethylaminopropyl) carbodiimide (EDC) and N-hydroxysuccinimide (NHS) was used to maintain consistency in the chemical composition between the rHCI and rHCIII matrices. Our rHC matrices are also glycosylated, containing the glycosaminoglycan chondroitin sulfate C (Fig. 1a). The hydrogels are then cross-linked in situ and assembled into a 3D matrix to provide a biomimetic niche for promoting endogenous repair within the infarcted myocardium.

Once cross-linked, the resulting rHCI and rHCIII matrices had equivalent denaturation temperatures of >45 °C (Fig. 1b). Similarly, water content of both rHC hydrogels was determined to be ≈94% ($p = 0.47$). The rHCI matrices were degraded ≈4 times more quickly than the rHCIII matrices upon exposure to collagenase (Fig. 1c), and also had 2.9-fold lower viscosity (Fig. 1d). Porosity is critical for cell infiltration and engraftment, and pore sizes of 9.3 ± 0.3 and 29 ± 1.0 μm were measured for the rHCI and rHCIII matrices, respectively (Fig. 1e). The presence of chondroitin sulfate did not affect either the denaturation temperature or the porosity of the material (Supplementary Fig. 2A, B). To characterize the injection and retention properties of the collagen matrices, they were labeled with Alexa-Fluor®594-NHS dye prior to injection to the MI mouse heart. Ex vivo fluorescence imaging revealed that the injected rHCI and rHCIII matrices were localized and retained within the injection site for at least up to 7 and 2 days, respectively. This was also confirmed by histological analysis of the infarct region using the labeled matrix fluorescence emission (Fig. 1f). Based on these labeling techniques, the injected rHC matrices appear to localize primarily within the epicardial tissue spreading across the infarct and border zone areas. This was also observed in fluorescent imaging and H&E staining of tissue sections at an earlier 2 h post-delivery time point (Supplementary Fig. 3A, B). Sections at different depths from the apex to the base of the rHC-treated hearts further confirm the presence of the rHC material within the epicardium and its spread over the infarct and border zone areas at 2 h post injection (Supplementary Fig. 3C, D).

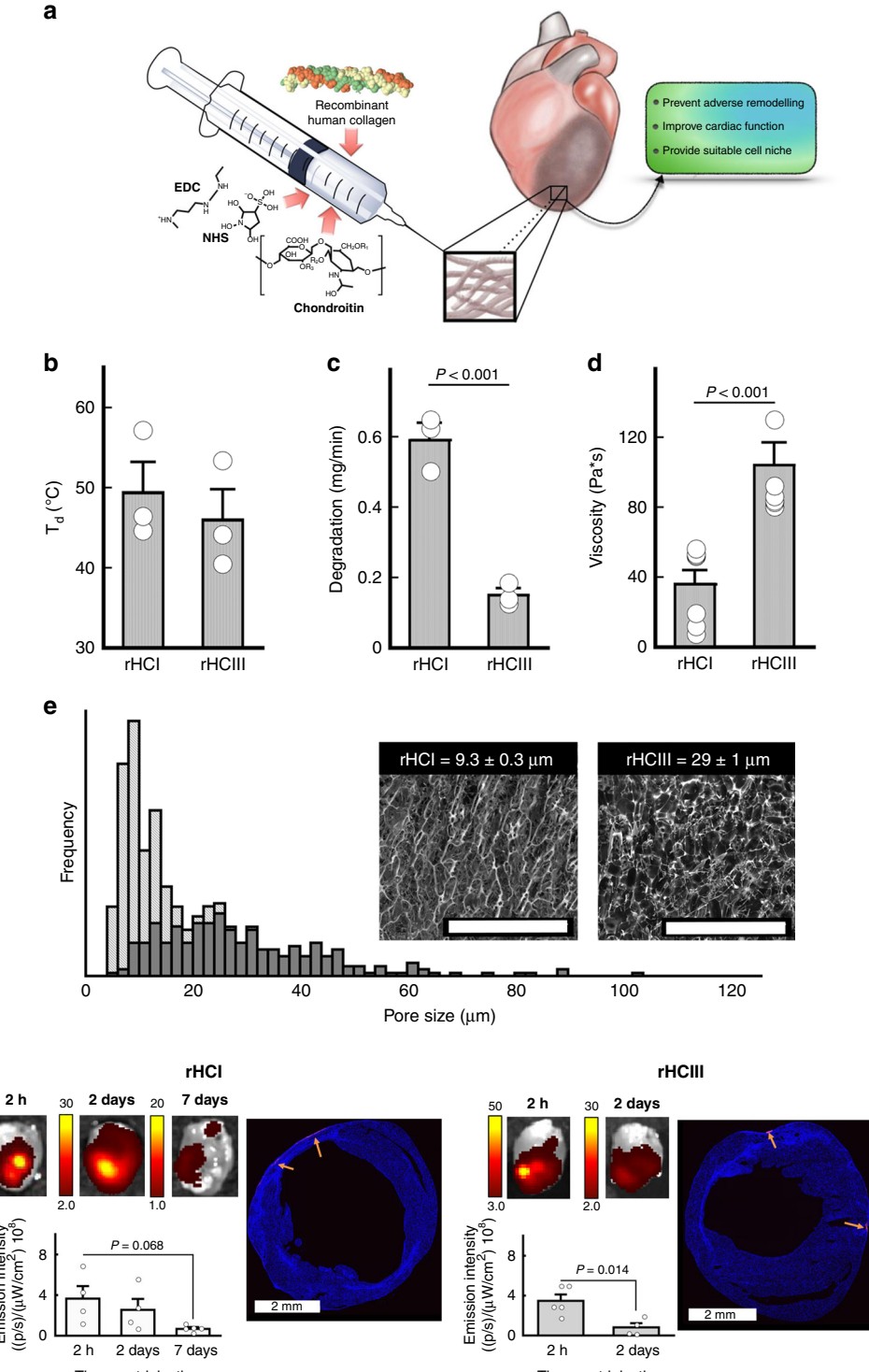

**Fig. 1** Preparation and physical characterization of injectable rHC matrices. **a** Schematic depicting the components, delivery, and reparative properties of the injectable material presented in this study. **b** Denaturation temperatures ($T_d$) (°C; $n = 3$), (**c**) in vitro degradation in collagenase (mg/min; $n = 3$), and (**d**) viscosity measurements (Pa·s; $n = 7$) for rHCI and rHCIII matrices. **e** Pore size distribution for the rHCI (gray bars) and rHCIII (black bars) matrices calculated from 250 individual pores per sample. Inset: Representative Cryo-SEM images of the rHC matrices. Scale bar = 200 μm. **f** Representative images of ex vivo imaging of MI mouse hearts at 2 h, 2 days, or 7 days, after treatment with the rHC matrices, labeled using Alexa-Fluor®594-NHS dye. Bottom histograms display the average total fluorescence emission for the different time points ($n = 4$). Also shown are representative histological sections of the myocardium at 2 days post injection. Arrows indicate representative regions containing the collagen matrices labeled with the fluorescent dye. $P$-values were determined by a two-tailed $t$ test. For **b–d**, ± data are presented as the mean ± SD and in **f** ± corresponds to SEM. Source data are provided as a Source Data file. For **b–e**, $n$ indicates number of hydrogel batches. For **f**, $n$ is the number of mice per group

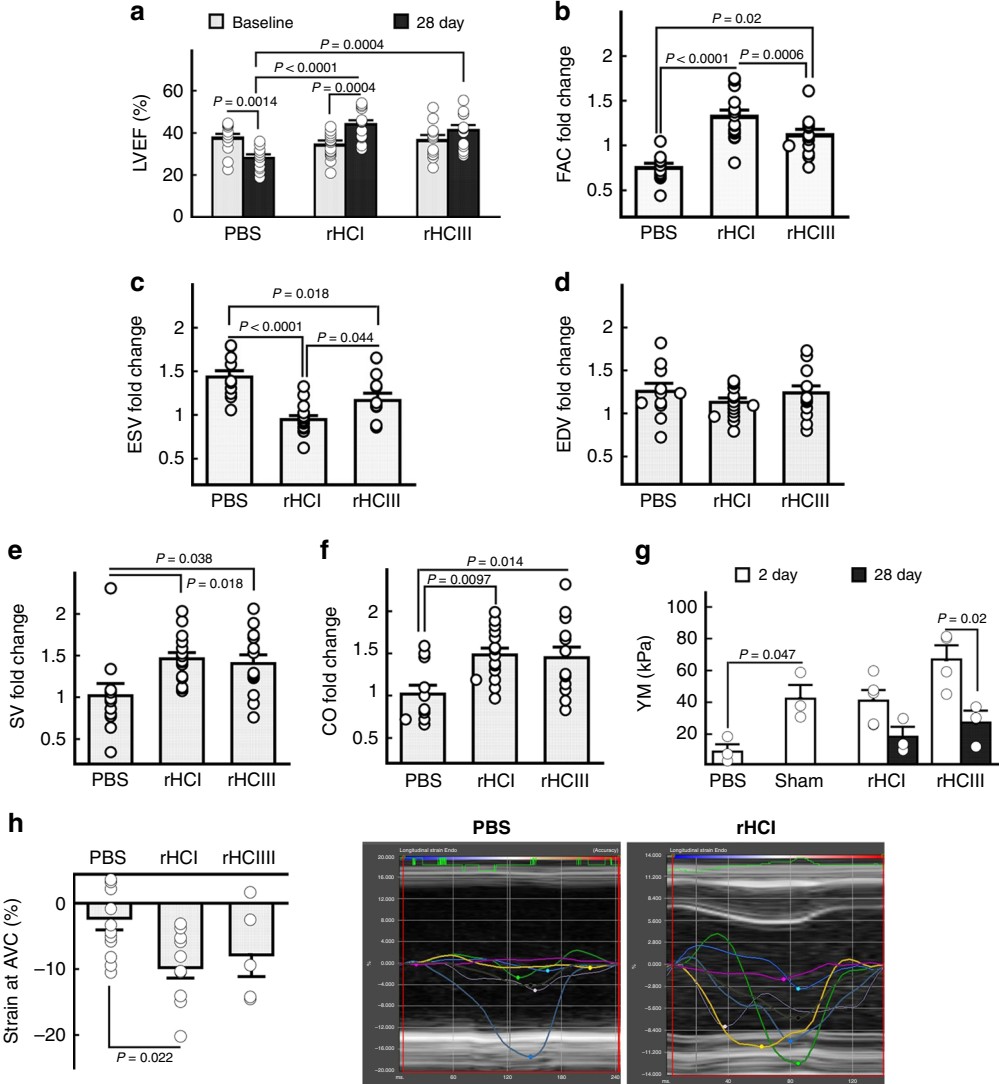

**Fig. 2** rHC matrices improve cardiac function post-MI and restore mechanical properties. **a** Left ventricular ejection fraction (LVEF) at baseline (time of treatment delivery) and at 28 days post treatment ($n = 11$ for PBS, $n = 15$ for rHCI, and $n = 13$ for rHCIII). Significance determined by a two-way ANOVA followed by Holm's correction for multiple comparisons. **b–f** The change in (**b**) fractional area change (FAC), (**c**) end-systolic volume (ESV), (**d**) end-diastolic volume (EDV), (**e**) stroke volume (SV), and (**f**) cardiac output (CO) between baseline and 28 days post-treatment ($n = 11$ for PBS, $n = 15$ for rHCI, and $n = 13$ for rHCIII). *P*-values were determined by an ANOVA followed by Holm-corrected multiple comparisons. **g** Young's modulus (YM) for scar/border zone tissue measured at 2 and 28 days ($n = 3–4$). **h** Left: Strain force reached at the aortic valve closure within the mid anterior LV wall at 2 days post injection for the different experimental groups ($n = 5–12$). Right: Representative longitudinal strain curves for the PBS and rHCI groups. The yellow traces correspond to the anterior mid region. Significance determined by a one-way ANOVA followed by Dunnett's comparison as well as a two-tailed Student's *t* test within a treatment group over time. The data are presented as the mean ± SEM. Source data are provided as a Source Data file. For **a–h**, *n* indicates the number of mice per group

In summary, the characterization of the rHC materials indicate that they possess suitable physical properties (denaturation at higher than body temperature, degradability by collagenase and porous structure), and that they remain in the infarct area for at least 2 days post injection, making them good candidates as injectable materials for treating the infarcted myocardium.

**rHC matrix improves post-MI cardiac function and morphology.** The therapeutic potential of the rHC matrices was tested using a mouse model of MI with the primary end point being left ventricular ejection fraction (LVEF) 28 days after treatment. One week after MI, mice were randomized to receive intramyocardial injection of rHCI matrix, rHCIII matrix, or PBS (control). Treatment with rHCI resulted in an improvement in

LVEF from 34.6 ± 1.6% at baseline to 44.2 ± 1.9% at 28 days post treatment; whereas LVEF between baseline and follow-up was unchanged for rHCIII-treated hearts, and it deteriorated in hearts that received PBS (Fig. 2a). At 28 days, LVEF was greater in hearts treated with rHCI or rHCIII matrix compared with PBS-treated hearts (Fig. 2a). Similar to PBS treatment, the injection of pure non-cross-linked collagens (rHCI or rHCIII) resulted in a reduction in LVEF (Supplementary Fig. 4), which indicates that rHCI and rHCIII require assembly into a cross-linked 3D matrix to confer their positive effects on post-MI cardiac function.

For other parameters of cardiac function, the fractional area change (FAC) at 28 days relative to baseline was superior in rHCI-treated hearts compared with PBS and rHCIII treatment (Fig. 2b). The change in end-systolic volume (ESV) was reduced in rHCI-treated hearts compared with the other 2 groups

(Fig. 2c), whereas no difference was observed for end-diastolic volume (EDV; Fig. 2d). ESV at 28 days was increased in the PBS group compared with rHCI matrix-treated mice (Supplementary Fig. 5A), indicating worse remodeling and a worsening of cardiac function in the PBS-treated mice. For EDV at 28 days, no difference was observed between groups, but it was significantly increased for rHCIII-treated hearts at 28 days compared with its baseline (Supplementary Fig. 5B). Also, both the rHCI and rHCIII treatments improved the fold change in stroke volume (SV) and cardiac output (CO) from baseline to follow-up vs. PBS-treated hearts (Fig. 2e, f). Notably, the tensile elasticity of the infarcted myocardium was restored by rHCI and rHCIII treatment at 2 days post injection to levels comparable with that of the healthy myocardium, and this was maintained up to 28 days for rHCI (Fig. 2g). In contrast, elasticity of the PBS-treated infarcted myocardium was severely compromised after 2 days, and was too weak to undergo testing at 28 days due to the extreme thinning and frailty of the ventricular wall. In vivo, analysis of longitudinal endocardial strain through speckle tracking echocardiography[41] demonstrated a significant improvement in the strain reached by the mid anterior LV wall at end systole, which is marked by the aortic valve closure (AVC), 2 days after injection of rHCI (Fig. 2h). The mid anterior LV wall is the segment of the myocardium targeted for hydrogel injection, as it contains the accessible infarct border zone. The longitudinal endocardial strain becomes more negative during systole as the heart shortens in this direction due to the stress placed on the myocardium during contraction. In healthy animals, strain should peak at the AVC, which is an indicator of end systole and strain at this point is a measurement of myocardial contractility. Therefore, the strain analysis provides evidence that the rHCI injection, but not the rHCIII, improves contractility in the border zone area of the LV wall where it was injected as compared with PBS-treated animals. Neither rHC matrix treatment affected the heart rate or any of the electrocardiographic parameters at 2 days post injection (Supplementary Table 1), with the exception of the PR interval for the rHCI matrix group. This indicates that the rHC matrices do not negatively interfere with electrical conductivity in the myocardium.

Cardiac remodeling can lead to enlargement of the heart in advanced stages post-MI. While mice that received PBS treatment had enlarged hearts, this was prevented in mice whose hearts were treated with rHCI matrix (Fig. 3a, b). Coinciding with this reduced remodeling, rHCI-treated hearts had greater remote ventricular wall thickness compared with the wall thinning observed with PBS treatment (Fig. 3c, d). Masson's trichrome staining at 28 days showed that scar size was reduced by 37 and 31% for the rHCI and rHCIII treatment groups, respectively, compared with PBS-treated hearts (Fig. 3c, e).

**Benefits of rHCI matrix on vascularity and cardiomyocytes**. To provide insight into the mechanisms underlying the functional benefit of rHC matrix therapy, immunohistochemistry was performed to assess vascularity and cardiomyocyte preservation in the scar and border zone at 28 days post treatment. Although no difference in vascular density (arterioles or capillaries) was observed in the scar area between groups, the number of capillaries was increased in the border zone of the hearts treated with rHCI or rHCIII matrix compared with PBS (Fig. 4a). The area of cardiac troponin I expression (a cardiomyocyte marker) was also shown to be greater in the border zone of hearts treated with rHCI matrix compared with PBS treatment (Fig. 4b), suggesting greater preservation of cardiomyocytes. Notably, the level of connexin 43 expression in the remote zone

was greater for the rHCI matrix group compared with PBS (Fig. 4c).

**rHC matrix treatment alters mononuclear cell recruitment**. Cell death post-MI stimulates an inflammatory response in the myocardium. Promoting the polarization of pro-wound healing M2 macrophages over the pro-inflammatory M1 macrophage phenotype has been shown to attenuate chronic inflammation and improve cardiac repair post-MI[42]. In our study, 4 weeks after rHCI matrix treatment there was a ≈1.5-fold increase in the number of $CD206^+$ M2 macrophages in the scar region compared with the PBS group (Fig. 5a). In response to death signals post-MI, inflammatory cells home to the infarcted myocardium from the circulation after their mobilization from the bone marrow. Therefore, we assessed the effect of rHC treatment on the homing of bone marrow cells to the myocardium using Cx3cr1-EGFP mice, whose bone marrow cells express green fluorescent protein (GFP). Our results showed a reduced total number of recruited $GFP^+$ cells in the left ventricle of hearts 2 days after treatment with rHCI matrix vs. those treated with rHCIII or PBS (Fig. 5b). By co-staining with inflammatory cell markers, we observed that the number of recruited $GFP^+CD38^+$ and $GFP^+CD11b^+$ leukocytes was reduced in rHCI-treated hearts (Fig. 5b).

For further characterization of mobilized bone marrow monocytes, we evaluated peripheral blood mononuclear cells in wild-type mice 2 days after treatment. There was an increase in the overall number of circulating monocytes and a trend for increased numbers of $Ly6C^{hi}$ monocytes in the rHCIII-treated mice compared with rHCI (Fig. 6a). No change in the other types of blood monocytes was seen between groups (Supplementary Fig. 6). In the myocardium, the rHCIII matrix increased the number of $Ly6C^{hi}$ monocytes 2 days after treatment, compared with PBS and rHCI (Fig. 6b), whereas there was no overall change in the numbers of leukocytes, macrophages, $Ly6C^{lo}$ monocytes, neutrophils, T cells, or B cells (Supplementary Fig. 7). Finally, there was a higher retention of $Ly6C^{lo}$ monocytes in the spleen for the rHCIII group compared with PBS (Supplementary Fig. 8).

**In vitro assessment of rHC matrices**. We next explored the interaction of the rHC matrices with macrophages in vitro to gain insight into how the materials may be regulating inflammatory cell function and conferring therapeutic benefit in vivo.

Bone marrow-derived macrophages from mice were used to evaluate the effect of the rHC matrices on macrophage function. The adhesion of unstimulated M0 macrophages to the rHCI and rHCIII matrices was equivalent with (Fig. 7a) or without (Supplementary Fig. 9) the inclusion of chondroitin sulfate; however, the number of M0 macrophages to migrate through the rHCIII matrix was twofold greater than for the rHCI matrix (Fig. 7b). The effect of the rHC matrices on macrophage polarization was also assessed. The results indicate that both matrices promote the polarization of the M2 wound healing phenotype (Fig. 7c) compared with TCPS, with rHCIII promoting greater M2 macrophage differentiation than rHCI. Gene expression analysis for several ECM remodeling proteins including MMP1, MMP2, MMP9, TIMP1 and TIMP2, as well as the M2 phenotype marker Arg1 was performed on macrophages cultured on TCPS, rHCI and rHCIII. The expression of MMP1 was increased in macrophages cultured on the rHCIII matrices vs. TCPS, while Arg1 expression was increased in macrophages cultured on rHCI compared with TCPS (Fig. 7d). To mimic the presence of oxidative stress post-MI, cultured macrophages were exposed to hydrogen peroxide ($H_2O_2$); and the results indicate that cells cultured on the rHCI and rHCIII matrices were

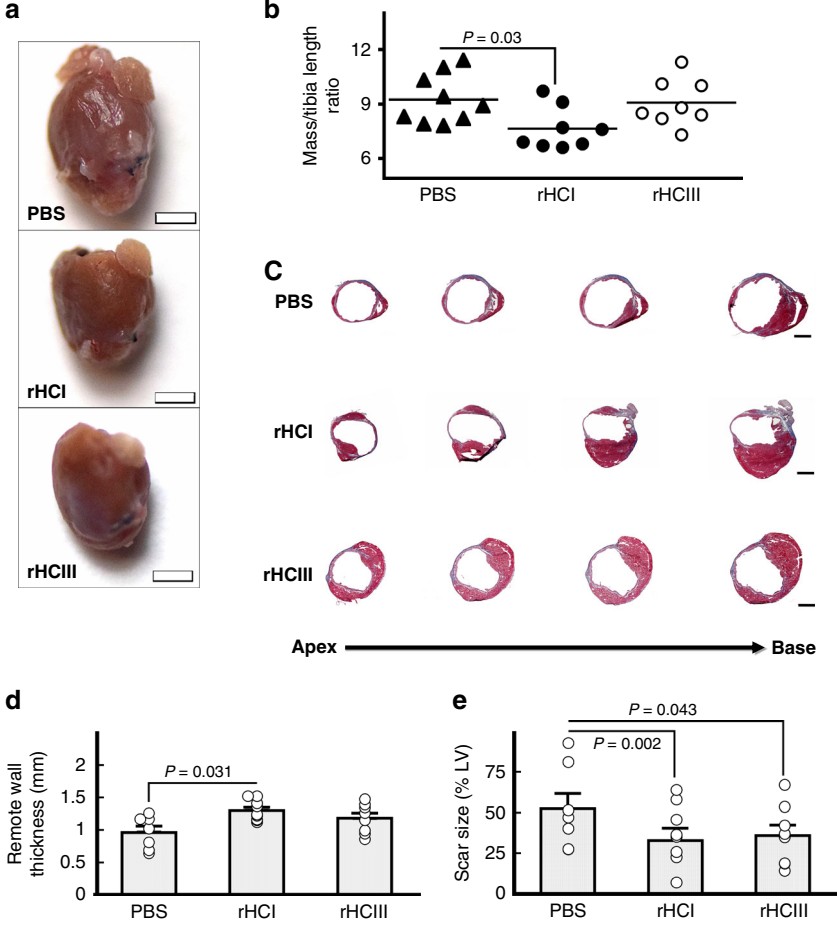

**Fig. 3** Morphology of MI hearts is improved by rHCI treatment. **a** Representative images of hearts harvested 28 days after treatment (bar = 2 mm). **b** Scatter plot for the heart mass/tibia length ratio measured 28 days post injection (*n* = 8–9). *P*-values were determined by a one-way ANOVA followed by Holm's multiple comparisons to PBS control. **c** Representative Masson's trichrome-stained sections of hearts harvested at 28 days (bar = 1 mm). **d** Wall thickness of the remote myocardium measured in Masson's trichrome-stained tissue sections at 28 days post treatment (*n* = 7–8). *P*-values were determined by one-way ANOVA using Holm's multiple comparison. **e** Scar size (% of LV) measured at 28 days post treatment in Masson's trichrome-stained tissue sections (*n* = 7–8). *P*-values were determined by multiple regression analysis, including treatment group (rHCI, rHCIII and PBS) and baseline LVEF. rHCI and rHCIII as compared with PBS were significant predictors of infarct size, in addition to baseline LVEF. The correlation coefficient for the model is 0.6. The data are presented as the mean ± SEM. Source data are provided as a Source Data file. For **a–e**, *n* indicates the number of mice per group

protected from $H_2O_2$-induced death (Fig. 7e). The same $H_2O_2$ exposure experiments were performed with NRVMs; no protective effect from the collagen matrices was observed (Fig. 7f).

## Discussion

Injectable hydrogels are attractive for cardiac applications due to their ease of use and the possibility of minimally invasive delivery. Therapeutically, they can provide physical stability to the infarcted myocardium, and can act as a biomimetic matrix for supporting cell function. Several biomaterials based on natural ECM proteins have been tested in animal models of MI and heart failure. Many have been shown to reduce adverse remodeling and limit the loss of function post-MI; but a material is needed that can increase cardiac function when used as a stand-alone therapy for treating myocardial scar at the late proliferative stage[43–45]. Furthermore, clinical translation of the materials tested so far can be limited and more challenging due to their animal origin[32,33].

We have assessed the performance of our therapy at the end of the proliferative phase post infarction (i.e., 7 days in mice, which is the equivalent of 14 days post-MI in humans)[11,38–40,46]. This time point represents a clinical opportunity to intervene to limit pathological ECM remodeling and promote infarct repair in

patients who have not responded to surgical interventions and other therapeutics. Using the two most abundant types of collagen found in the healthy heart, we developed injectable rHCI and rHCIII matrices. The rHCI and rHCIII matrices exhibited differences in their biophysical properties (viscosity, porosity, and enzymatic degradation), but both demonstrated good injectability and gelation at 37 °C, making them suitable for delivery to the heart that has sustained an MI. Although injecting collagen post-MI may seem counterintuitive given the presence of the collagenous scar, the composition and mechanical properties of the scar are vastly different from the normal myocardium[10,15]. Supporting their use as a therapy post-MI, (animal-derived) collagen materials have been shown to improve angiogenesis and tissue integration, reduce inflammation and apoptosis, and limit negative remodeling and the loss of cardiac function[24,26–31,47–50]. Furthermore, the safety and efficacy of a porcine myocardial ECM hydrogel, composed primarily of collagens, was demonstrated in a pre-clinical pig MI model[20].

In this study, improved cardiac function between baseline (i.e., time of treatment delivery) and 4 weeks post treatment was observed only in mice treated with the rHCI matrix; whereas cardiac function was preserved with rHCIII treatment, and worsened in the control group. The improvement in LVEF for the

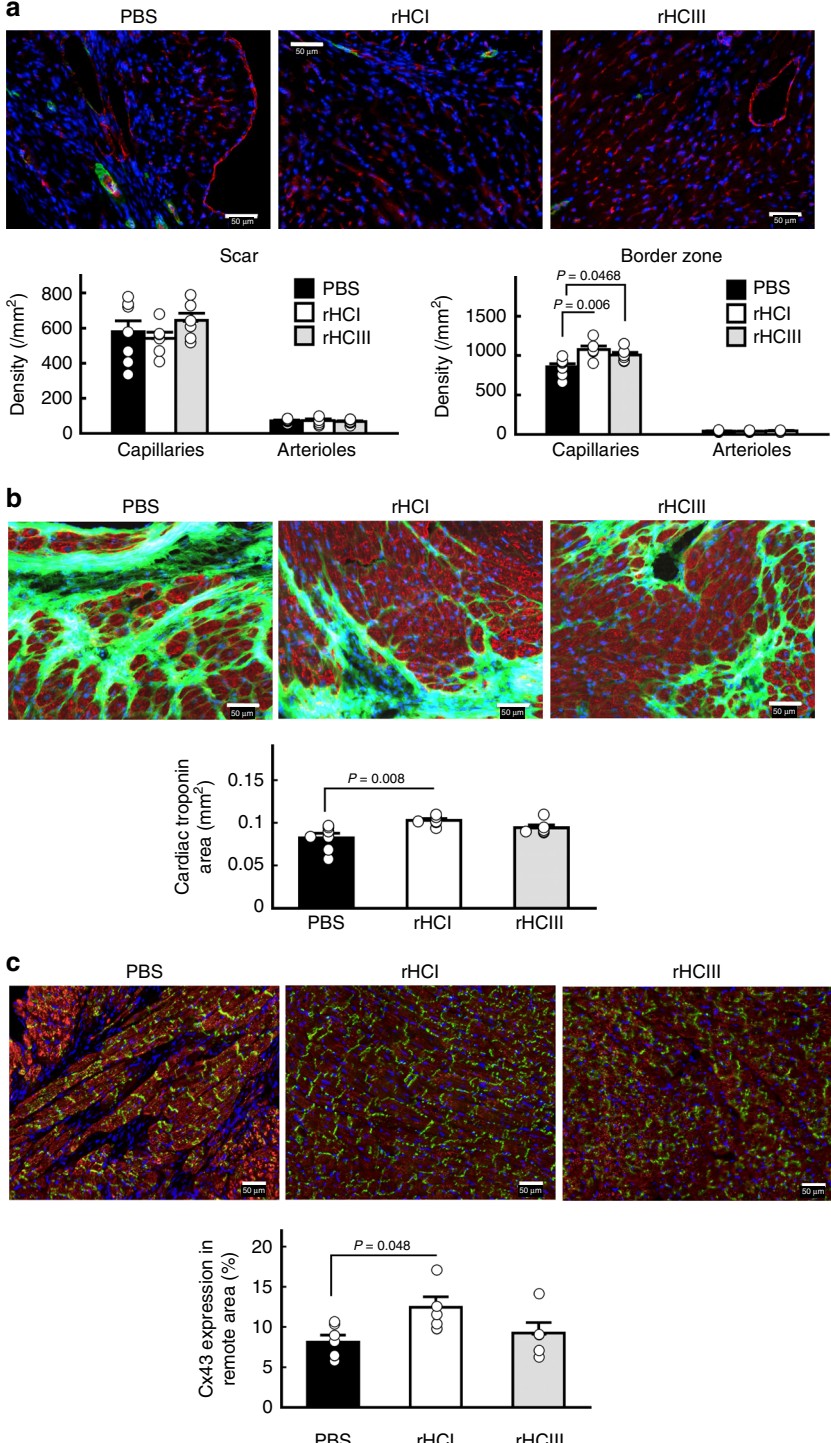

**Fig. 4** rHC matrix increases vascularity and preserves cardiomyocytes post-MI. **a** Immunohistochemistry analysis and quantification (number/mm$^2$) of capillary and arteriole number in the myocardial scar and border zone after treatment with PBS, rHCI, or rHCIII. CD31, red; α-SMA, green; DAPI, blue ($n =$ 6–7). **b** Immunohistochemistry analysis and quantification of total area (in mm$^2$) positive for cardiac troponin I (cTnI) staining in the border zone after treatment with PBS, rHCI, or rHCIII. cTnI, red; wheat germ agglutinin, green; DAPI, blue. **c** Immunohistochemistry analysis and quantification of connexin 43 (Cx43) expression (percent area) in the remote area after treatment with PBS, rHCI, or rHCIII. Cx43, green; cTnI, red; DAPI, blue ($n =$ 6–7). *P*-values were determined by one-way ANOVA using Holm's multiple comparison. The data are presented as the mean ± SEM. Source data are provided as a Source Data file. For **a**–**c**, *n* indicates the number of mice per group

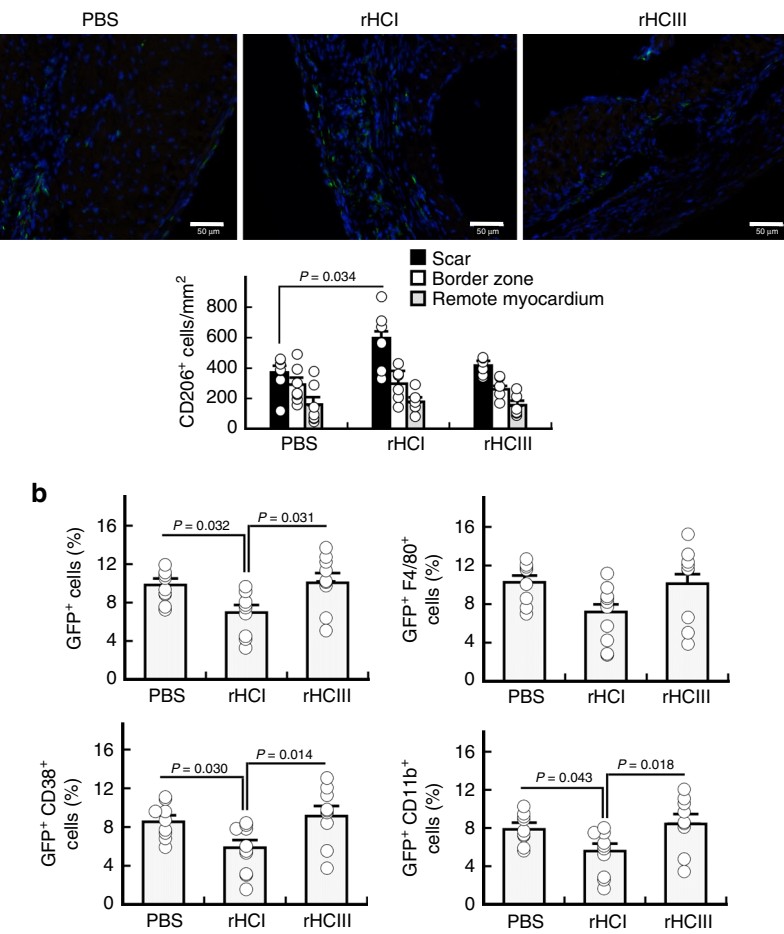

**Fig. 5** rHC matrix treatment alters bone marrow cell and macrophage response. **a** Immunohistochemistry analysis and quantification of the number of CD206[+] macrophages (per mm[2]) in the scar, border zone and remote myocardium of MI hearts treated with PBS, rHCI, or rHCIII. CD206, green; DAPI, blue ($n = 6$–7). **b** Flow-cytometry analysis for the number of GFP[+], GFP[+]F4/80[+], GFP[+]CD38[+], and GFP[+]CD11b[+] cells recruited from the bone marrow to the left ventricle of Cxcr3-EGFP MI mice 2 days after PBS, rHCI, or rHCIII treatment ($n = 9$–10). *P*-values were determined by one-way ANOVA using Holm's multiple comparison. The data are presented as the mean ± SEM. Source data are provided as a Source Data file. For **a**, **b**, $n$ indicates the number of mice per group

rHCI group was similar to that observed in MI mice that received rat-tail collagen matrix treatment delivered acutely at 3 h post-MI[26]. The recovery of LVEF function was accompanied by reduced end-systolic volume (ESV), as well as improved fractional area change, stroke volume, and cardiac output, which are all indicators of overall improvement in cardiac function. In contrast, end-diastolic volume (EDV) was not improved in hearts that received rHC treatment. However, this prognostic indicator is typically used in assessing adverse remodeling months after MI has occurred (4–6 months in humans; equivalent to ~2–3 months in mice), and our furthest time point evaluated is at 5 weeks post-MI (4 weeks post treatment). Furthermore, while the use of EDV alone as a clinical end-point has been questioned, ESV has been shown to be an early predictor for recovery of the MI heart[51–55].

The mechanical properties of the myocardium, measured ex vivo (tensile strength), were comparably restored within days of receiving rHCI or rHCIII treatment, but only rHCI treatment improved the longitudinal endocardial strain measured in vivo (strain at the aortic valve closure), and both rHC treatments were accompanied by a decrease in the scar size. Notably, rHCI-treated hearts were smaller, and the thickness of the remote myocardium was greater in the rHCI group. This suggests that rHCI treatment can reduce dilation and adverse ventricular remodeling, which has been linked to superior clinical outcomes[56]. Thus, the

capacity of our materials to improve cardiac function post-MI after the scar has formed (late proliferative phase), particularly the rHCI formulation, presents a desirable and clinically translatable platform for treating MI patients beyond the initial inflammatory phase[44].

Vascularization of the infarct and peri-infarct regions after MI is critical to limiting necrosis and preserving cardiac function[57,58]. Although no differences in vascular density (arterioles or capillaries) were observed in the scar region between treatment groups; the capillary density was greater in the border zone for both rHCI and rHCIII groups. Increasing the microvasculature in the peripheral area of the infarct can lead to salvage of surrounding myocardium, limit expansion of the infarct and prevent the progression to heart failure[59]. Modulating inflammation and macrophage phenotype has also been shown to limit adverse remodeling in the MI heart[60,61], and is a possible mechanism that may be linked to the observed benefits of rHCI matrix treatment. At 28 days post-MI, the number of M2 macrophages in the scar region was ≈1.5-fold greater for rHCI vs. PBS treatment. Furthermore, experiments using Cx3cr1-EGFP mice showed that the rHCI matrix reduced the total number of GFP[+] monocytes that were recruited from the marrow to the infarct site, 2 days after treatment delivery (9 days post-MI). Therefore, it appears that the rHCI matrix can reduce inflammation early on and promote

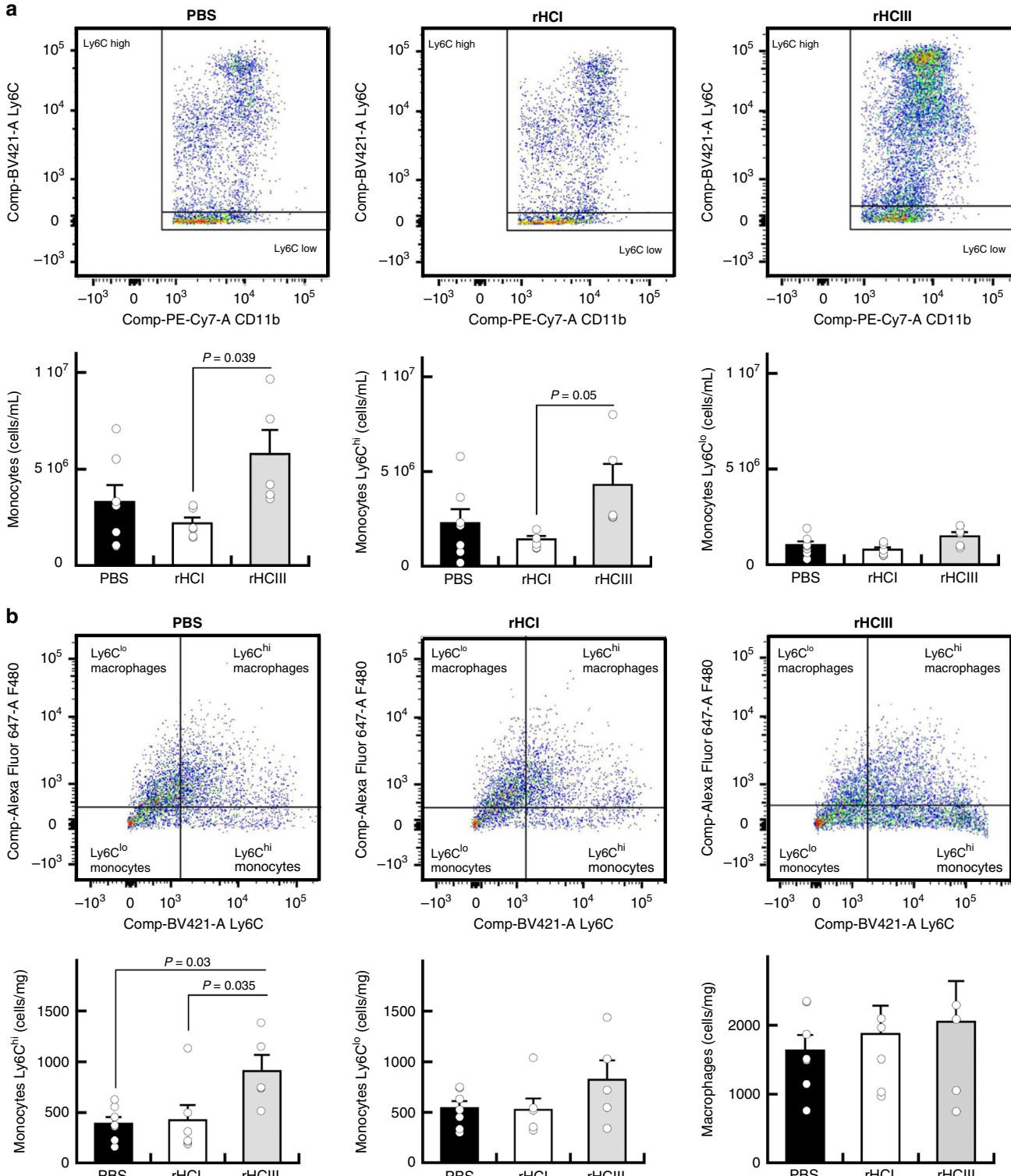

**Fig. 6** rHCIII matrix increases Ly6C$^{hi}$ monocyte mobilization and accumulation in the heart. Flow-cytometry plots show F480 and Ly6C expression of mononuclear cells (CD45$^{+}$CD11b$^{+}$Ly6G$^{-}$) in the blood (**a**) and heart (**b**) of mice at 2 days post treatment. Macrophages were considered F480$^{+}$, while monocytes were F480$^{-}$ and classified based on the expression of the Ly6C marker. The data are presented as the mean ± SEM ($n = 5$–7). $P$-values were determined by a one-way ANOVA for effect of treatment followed by Holm's correction for multiple comparisons. Source data are provided as a Source Data file. For **a**, **b**, $n$ indicates the number of mice per group

wound healing in the long-term. This may be an explanation for the increased expression of cardiac troponin I in the border zone and Cx43 in the remote region of rHCI-treated hearts compared with either PBS or rHCIII. The analysis of monocyte populations

2 days post treatment in wild-type mice showed an increase in the mobilization of monocytes into the blood and greater accumulation of Ly6C$^{hi}$ cells in the myocardium for the rHCIII group. Also, an increased number of Ly6C$^{lo}$ cells was seen in the spleen

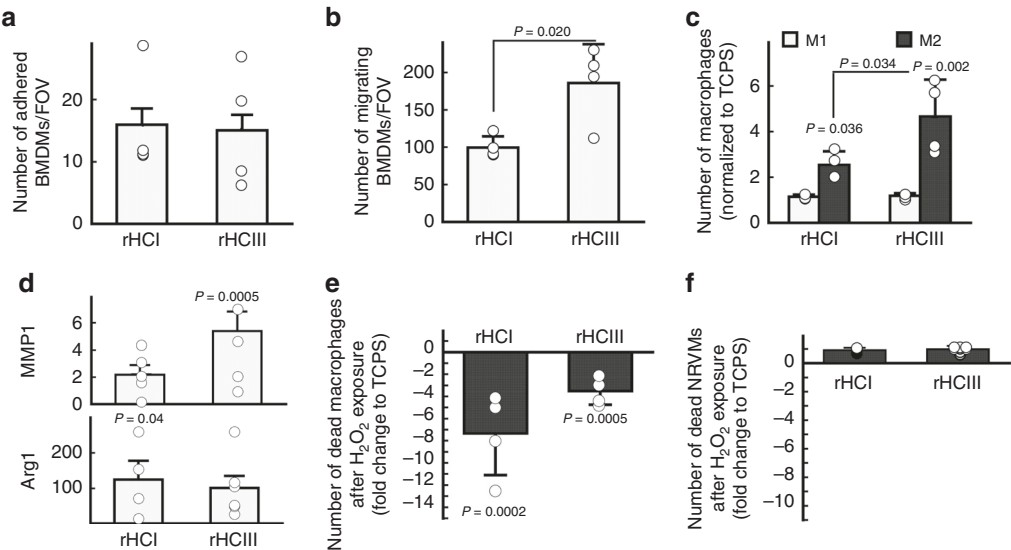

**Fig. 7** rHC matrices support wound-healing macrophages. **a** The number of adherent macrophages after culture on rHC matrices ($n = 4$). **b** Number of macrophages that migrated through 300 μm thickness rHC matrices ($n = 4$). **c** M1 and M2 macrophage polarization determined by CD86 and CD206 expression, respectively, after culture on rHC matrices ($n = 4$). **d** MMP1 and Arg1 mRNA expression in macrophages cultured on rHC matrices for 72 h relative to TCPS ($n = 4$). **e** Percentage of dead macrophages (7-AAD$^+$ cells) after 3 h incubation with $H_2O_2$ ($n = 4$). **f** Percentage of dead neonatal rat ventricular myocytes (NRVMs) after 3 h incubation with $H_2O_2$ ($n = 4$). $P$-values were determined by a one-way ANOVA using Holm's multiple comparison (**c**, **d**, **e**, and **f**), or by a two-tailed $t$ test (**a** and **b**). All data are presented as the mean ± SD. Source data are provided as a Source Data file. For **a**–**e**, $n$ indicates the number of mice (mouse bone marrow) per group. For **f**, $n$ indicates the number of hydrogel batches

for the rHCIII group, suggesting some degree of inflammatory response to the rHCIII matrix, but not the rHCI matrix. Altogether, the effects of the rHCI matrix on vascularity, anti-inflammatory/pro-wound healing, and cardiomyocyte viability may play a role in its ability to improve cardiac function in the MI mouse heart. To gain further insights on the biological activity of the matrices, we performed in vitro experiments.

Upon injection, monocytes/macrophages will be recruited to the rHC matrices, and cell adhesion will dictate the ensuing initial response. In vitro experiments showed no difference in the adhesion of murine BMDMs between the two rHC materials; however, BMDM invasion into the rHCIII matrix was twofold greater compared with the rHCI matrix. Since the pore size for rHCIII was considerably larger than that of rHCI, greater cell permeation for rHCIII is not surprising. Both rHC matrices promoted greater M2 polarization and expression of M2 markers (MMP1 and Arg1) than TCPS, but more so with the rHCIII matrix than the rHCI matrix. This is in contrast to our in vivo observations, in which only the rHCI matrix, but not rHCIII, led to greater numbers of M2 macrophages in the infarcted heart long-term compared with PBS control treatment. This may be related to differences between the in vitro and in vivo conditions. For example, it has been shown that M2 polarization is stimulated to a greater extent in 3D vs. 2D environments[62]. Thus, it is plausible that, although cells are seeded on the surface of the materials in vitro (i.e., in 2D), the larger pore size of the rHCIII material may result in more cells being exposed to a 3D-like environment leading to greater M2 macrophage polarization. In the in vivo setting, the injected material is subject to enzymatic degradation, which was shown to be greater for the rHCIII material; thus, we speculate that the slower clearance of rHCI leads to an increased number of M2 macrophages over time in the rHCI treatment group.

Oxidative stress post-MI is considered to be a major factor in cardiac remodeling[63]. We therefore sought to examine the protective effect of the matrix against oxidative stress-induced death of macrophages (BMDMs), which were increased in number

within the myocardium of rHCI-treated hearts. Both rHCI and rHCIII matrices were able to reduce the number of dead BMDMs in the presence of hydrogen peroxide. However, the NRVMs remain susceptible to $H_2O_2$-induced death when cultured on the rHC matrices. This suggests that our rHC materials may act as a protective niche limiting the loss of wound-healing macrophages in conditions of oxidative stress.

In summary, we have demonstrated that our rHCI matrix made from clinical grade human collagen is able to prevent adverse cardiac remodeling, and to improve cardiac function in the MI heart when applied during the late proliferative phase. Our results are the first step in the development of a stand-alone collagen biomaterial therapy that is clinically translatable. In terms of the potential cost savings from our new therapy, it should be considered that in Canada alone there are 600,000 patients living with advanced heart failure, whose healthcare costs amounts to more than $2.8B every year[64]. These patients require long and frequent hospital stays. Thus, a therapy that is able to reduce the progression to clinical heart failure would greatly improve both quality and quantity of life for many MI patients and potentially save millions in healthcare spending. We expect these findings will pave the way for providing a biomaterial therapy suitable for combination with cell therapies or surgical procedures aimed at restoring cardiac function in MI patients.

## Methods

**Study design.** Here, we conducted a study to (i) design clinically relevant collagen hydrogels using recombinant human collagens type I and type III; (ii) characterize and compare the physical properties of the rHCI and rHCIII materials; (iii) evaluate the therapeutic potential of the materials for treating MI in mice; and (iv) identify mechanisms underlying the observed therapeutic effects of rHC treatment. For in vivo experiments, LAD artery ligation in mice was chosen as a clinically relevant and well-established animal model of MI. For functional, histological, and molecular evaluations, the number of animals per group was minimized to three to fifteen mice, as specified in the figure legends. Mice were randomly assigned to treatment groups and all analyses were performed in a blinded fashion.

**Preparation and characterization of rHC matrices.** A 1% collagen solution was prepared by dissolving 0.1 g of lyophilized collagen (rHCI and rHCIII, from

Fibrogen) in 10 ml of ultra-pure $ddH_2O$. Chondroitin sulfate (CS; Wako), N-ethyl-N-(3-dimethylaminopropyl) carbodiimide (EDC) and N-hydroxysuccinimide (NHS) were added to produce a final mixture with a mass ratio of 1:4:0.5:0.3 for collagen:CS:NHS:EDC. The materials were prepared on ice using an enclosed system that allows homogenous mixing without adding bubbles. After thorough mixing, the pH was adjusted to 7.4 by NaOH (1.0 N). Matrices without CS were prepared similarly but with PBS added to compensate for the volume of CS. Matrices labeled with Alexa-Fluor®594-NHS were prepared by adding 20 µL of the dye stock solution (1 mg/mL in DMSO) to the gels before adding the NaOH (25 nmol of dye per hydrogel), followed by 20 mixing steps.

**Material viscosity.** Viscosity measurements were carried out using a Brookfield R/S plus rheometer (Brookfield) at 37 °C. A C25–2/30 conical spindle was used to compress the material (50 µm displacement) onto a temperature controlled pedestal. Viscosity was measured with a ramp rotational block at a speed of 1/min units pre-set at a shear rate of five units over a time of 30 min.

**Degree of cross-linking.** Differential scanning calorimetry (DSC) experiments were carried out to assess the degree of cross-linking. Briefly, measurements were performed in a Q2000 differential scanning calorimeter (TA Instruments) in the range of 8 to 80 °C using a scan rate of 5 °C $min^{-1}$. Collagen matrices (5–20 mg) were surface-dried with filter paper and hermetically sealed with an aluminum lid (Tzero; TA Instruments) in an aluminum sample pan (Tzero; TA Instruments). The denaturation temperature ($T_d$) was measured at the onset of the endothermic peak.

**Water content.** Water content of the materials was measured by weighing the wet weight ($W_0$) of the sample, equilibrated in PBS for 96 h at 4 °C. The material was then vacuum-dried at room temperature for 96 h to obtain the dry mass ($W$). The total water content of the hydrogels ($W_t$) was then calculated according to the equation:

$$W_t = \frac{(W_0 - W)}{W_0} \times 100 \tag{1}$$

**Degradation.** Enzymatic degradation was measured using 50–100 mg of hydrogel placed in 5 ml of type I collagenase (10 U/ml in PBS) at 37 °C. The remaining solid mass was measured over a period of up to 24 h. The degradation rate is calculated from the initial slope of the plots of remaining mass vs. time and reported as mg/min.

**Material porosity.** Low-temperature scanning electron microscopy (Cryo-SEM) measurements were performed at −50 °C using a Tescan (Model: Vega II – XMU) equipped with a cold-stage sample holder, a backscatter electron detector (BSE), and a secondary electron detector (SE). Pore sizes were measured from at least 250 individual from 4 to 6 random areas of the sample using ImageJ® software. Diameter for pores were quantified using the longitudinal axis of the pore using the straight line tool. Area of the image was adjusted against the size scale bar obtained from the Tescan Vega II system[65].

**Animal experiments.** All procedures were approved by the University of Ottawa Animal Care Committee, and performed according to the National Institute of Health Guide for the Care and Use of Laboratory Animals.

**MI model.** MI was induced in 9-week-old female C57BL/6 mice (Charles River; number of mice/group are in the figure legends) and treatment delivery was performed using an established protocol[26,27]. Mice were anesthetized (2% isoflurane), intubated, and the heart was exposed via fourth intercostal thoracotomy. The left anterior descending coronary artery (LAD) was then ligated just below its emergence from the left atrium. This procedure results in a large MI involving the anterolateral, posterior, and apical parts of the heart, which was confirmed at the time of surgery by myocardial blanching in the region supplied by the artery. Short-acting buprenorphine was administered at least an hour prior to surgery, and long-acting buprenorphine was administered subcutaneously immediately before surgery for perioperative analgesia. At 1-week post-MI (baseline), mice were randomly assigned to receive treatment of PBS (control), rHCI, or rHCIII matrices, delivered in five equivolumetric intramyocardial injections (10 µl each site, 50 µl total) through a 27-G needle using an ultrasound-guided closed-chest procedure. The syringe is secured in a micromanipulator (VisualSonics), and prior to the injection procedure, both the needle and RMV scanhead probe are aligned along the heart long-axis. Via the ultrasound field-of-view, the micromanipulator is used to position the needle tip was in the desired location within the myocardium for delivery of the injections (see Supplementary Fig. 1 and Supplementary Video 1). Mice were killed by terminal anesthesia at 2 days or 4 weeks post-treatment and hearts were collected for histology and/or measurements of the mechanical properties.

**Echocardiography.** Transthoracic echocardiography was performed on long-axis views using a Vevo770 system in B mode with a 707B series real-time microvisualization scanhead probe (VisualSonics). The imaging was performed at baseline before treatment injection (7 days post-MI) and at 28 days post-injection to determine left ventricular ejection fraction (LVEF), fractional area change (FAC), end-systolic volume (ESV), end-diastolic volume (EDV), stroke volume, and cardiac output. Note that LVEF, ESV, and EDV are used as clinical predictors of HF and survival after MI[44].

**Ex vivo imaging of Alexa-Fluor®594-labeled matrix.** The chemically tagged rHC matrices (25 nmol of dye per hydrogel) were injected to the infarcted mouse heart, as described above. Animals were killed at 2 h, 2 days, and 7 days after treatment, and hearts were harvested and imaged ex vivo by IVIS® Spectrum (PerkinElmer) to visualize rHCI and rHCIII distribution within the hearts ($\lambda_{excitation}$: 570 nm; $\lambda_{emission}$: 640 nm). For histology, tissue sections were prepared in acetone and stained with DAPI followed by imaging using a Leica Aperio Versa slide scanner with a final magnification of ×20 and a Z-stack with eight steps at 1.5 µm.

**Strain analysis.** Transthoracic echocardiography was performed on long-axis views using a Vevo3100 system in B mode with a MX400 series real-time microvisualization scanhead probe (VisualSonics). The imaging was performed at baseline before treatment injection (7 days post-MI) and at 2 days post injection to determine longitudinal endocardial strain at the time of aortic valve closure (representing end systole). Strain in Segment 5 (the anterior mid segment), corresponding to the border zone area targeted for treatment injection, was analyzed using the Vevostrain application of Vevo LAB 3.1.1 software (VisualSonics)[41]. Representative examples for the measurements carried out are included in Supplementary Fig. 10 for all the experimental groups plus a healthy (noninfarcted) animal.

**Electrocardiography.** Electrocardiograms were obtained simultaneously with echocardiography using a Vevo3100 system (VisualSonics) at baseline before treatment injection (7 days post-MI) and at 2 days post injection. Electrocardiograms were exported from Vevo LAB 3.1.1. software (VisualSonics). The length of the PR, QT, and QRS intervals was determined using ImageJ software (Supplementary Table 1).

**Mechanical properties of the myocardium.** Mice were euthanized by terminal anesthesia at 2 or 28 days post treatment, and the hearts were harvested. Rectangular pieces ($2.5 \times 5$ mm) of the left ventricle comprising the scar and border zone were excised. To determine the tensile elasticity of the tissue (Young's modulus), the samples were subjected to mechanical testing in an Instron mechanical universal tester (Model 3342, Instron) equipped with Series IX/S software, using a crosshead speed of 10 mm $min^{-1}$.

**Histology/Immunohistochemistry.** Slides of myocardial tissue sections were prepared from a subset of hearts that were not used for mechanical properties measurements. At 28 days post injection, hearts were harvested, perfused with PBS, embedded in OCT, and snap-frozen in liquid nitrogen. Sections of 10 µm were cut using a cryostat. To assess scar size, tissue sections were fixed in 4% PFA for 1 h and stained with Masson's trichrome procedure (Sigma). Images taken with an Olympus BX50 microscope using a 2 × objective, and eight sections per mouse were used to measure remote wall thickness and to determine scar size using the mid-line arc method using MIQuant software[66]. For immunohistochemistry, tissue sections were fixed in acetone for 20 min, permeabilized with 0.1% Triton for 10 min and then blocked in 10% serum for 1 h at room temperature. Primary antibodies were applied overnight at 4 °C in 10% serum, and then slides were rinsed and treated with secondary antibodies for 1 h at room temperature, before being mounted with fluorescent mounting medium (Dako). For the detection of blood vessels and myofibroblasts, PECAM-1 (also known as CD31; Santa Cruz 101454, 1:50) and α-SMA (Abcam 5694, 1:200) antibodies were used and detected with AF594 anti-rat (Life Technologies A11008, 1:500) and AF488 anti-rabbit (Life Technologies A11007, 1:500), respectively. M2 macrophages were detected by AF488-conjugated anti-CD206 antibody (Biolegend 141710, 1:50). For cardiac troponin I staining, sections were incubated with AF488-labeled wheat germ agglutinin (ThermoFisher, W11261) for 1 h at 37 °C followed by incubation with goat cardiac troponin I primary antibody (Abcam ab56357, 1:200) and detected by donkey anti-goat AF594 secondary antibody (Life Technologies A-11058, 1:500). Finally, for connexin 43 staining, sections were incubated with connexin 43/GJA1 (Abcam ab11370, 1:400) and cardiac troponin I (Abcam ab188877, 1:200) primary antibodies followed by detection with AF488 anti-rabbit (Life Technologies A11007, 1:500) and donkey anti-goat AF594 secondary antibody (Life Technologies A-11058, 1:500), respectively. Fluorescent images were obtained using a Zeiss Axio Observer microscope with a ×20 objective and for the connexin 43 stained sections a Leica Aperio Versa slide scanner with a 20 × objective. For all IHC stains, four sections per mouse were analyzed and for each section 2–4 images within the border zone, infarct and remote regions were used for analysis. For H&E staining, tissue sections were fixed in 10% formalin. Sections were then stained first in hematoxylin gill's solution No. 2 for 7 min, followed by acid alcohol differentiation, and then 0.5% eosin for 7 min, followed by dehydration (all solutions from Sigma). Images were taken with an Olympus BX50 microscope. The border

zone was designated as the FOV directly adjacent to either side of the infarct region that begins when 50% of the LV is fibrotic scar tissue (see Supplementary Fig. 11 for a schematic depiction).

**Cx3cr1-EGFP mouse experiments**. To evaluate the recruitment of circulating mononuclear cells to the myocardium following rHC treatment, B6.129P-Cx3cr1 tm1Litt/J mice (Cx3cr1-EGFP) were purchased from the Jackson Laboratory. These mice express enhanced green fluorescent protein in monocytes, dendritic cells, NK cells, and brain microglia, and are a model reported for the study of mononuclear cell recruitment to the heart[67]. At 1-week post-MI, mice received treatment of 50 μl of PBS, rHCI, or rHCIII, as described above. Animals were killed 2 days after treatment, and blood was collected into EDTA tubes. Also, hearts were perfused with PBS and the right ventricle and the apical region of the left ventricle were collected. The tissues were rinsed with HBSS and digested in 2.4U/ml dispase I (Roche) and 1 mg/ml Collagenase B (Roche) for 40 min at 37 °C. Samples were washed three times with PBS, centrifuged for 5 min at 400 g, and the isolated cells were prepared for flow cytometry (FACS Aria III; Becton Dickinson). Cells were labeled with APC anti-mouse/human CD11b (Biolegend 101211), PE anti-mouse Ly-6G/6C (Biolegend 108407), PE/Cy5 anti-mouse F4/80 (Biolegend 123111), PE/Cy7 anti-mouse CD38 (Biolegend 102717) and Alexa Fluor® 700 anti-mouse CD206 (Biolegend 141733), following the manufacturer-recommended dilutions (0.25 μg/L per $1 \times 10^6$ cells for CD11b, Ly-6G/6C, CD38, and CD206, and 1 μg/L per $1 \times 10^6$ cells for F4/80). See Supplementary Fig. 12 for sorting/gating strategy.

**Cell isolation and flow cytometry for monocyte subsets**. Two days post-treatment injection mice were killed by CO₂ inhalation followed by cervical dislocation. Blood was collected from mice through cardiac puncture in a 50 mM EDTA solution, and RBCs were lysed with red blood cell (RBC) lysis buffer according to the manufacturer's protocol (Biolegend 420301). Cells were isolated from harvested mouse hearts using a digestion buffer containing: DNAse I (50 U/μL; Sigma D5025), collagenase type II (400 U/ml; ThermoFisher 17101015), collagenase D (0.15 U/mL; Sigma 11088866001), and hyaluronidase (10 U/mL; Sigma H3506). Following an hour digestion at 37 °C, isolated heart cells were passed through a 70 μm filter. Finally, harvested mouse spleens were mashed and triturated through a 70 μm filter followed by incubation with red blood cell (RBC) lysis buffer. Cell pellets were collected by centrifugation at ×400g for 5 min at 4 °C. Isolated cells from all tissues were incubated with Zombie Aqua fixable viability dye (1:500, Biolegend 423101) for 20 min at room temperature. Next, Fc receptors were blocked with TruStain X reagent (1:100, Biolegend 103319) for 10 min at room temperature. The cells were then incubated with an antibody cocktail for 45 min at room temperature containing CD45-APC/Fire 750 (1:160, Biolegend 30-F11), CD11b-PE/Cy7 (1:80, Biolegend M1/70), Ly6G-PerCP/Cy5.5 (1:160, Biolegend 1A8), CD3-PE (1:160, Biolegend 17A2), B220-AF488 (1:50, Biolegend RA3-6B2), F480-AF647 (1:100, Biolegend BM8) and Ly6C-BV421 (1:80, BD Biosciences AL-21). Flow cytometry was performed using a BD FACS Aria III, and data were analyzed with FlowJo V10.5.2 software (see Supplementary Fig. 12 for sorting/gating strategy). The total viable cell number for each tissue post isolation was determined by trypan blue exclusion using a hemocytometer. The total number of cells within each leukocyte population was determined using the percentage of live cells for a given cell population, and the total number of live cells counted post-isolation which was then normalized to the mg of tissue or mL of blood collected. Gating strategy: Single cells were gated based on FSC-A and FSC-H. Live single cells were gated on low staining for Zombie Aqua live/dead dye. From this live single cell population leukocytes were CD45⁺. Leukocyte subsets were characterized as follows: neutrophils CD45⁺CD11b⁺Ly6G⁺, Ly6C low monocytes CD45⁺CD11b⁺Ly6G⁻F480⁻Ly6C⁻, Ly6C high monocytes CD45⁺CD11b⁺Ly6G⁻F480⁻Ly6C⁺, macrophages CD45⁺CD11b⁺Ly6G⁻F480⁺, T cells CD45⁺CD11b⁻Ly6G⁻CD3⁺B220⁻ and B cells CD45⁺CD11b⁻Ly6G⁻CD3⁻B220⁺. Note for blood F480 expression was not used in the gating strategy. Gates were set relative to isotype controls.

**Neonatal cardiomyocyte studies**. Neonatal rat ventricular myocytes (NRVMs) were freshly isolated using an established protocol[68]. Left ventricles collected from 2-day-old Sprague–Dawley rats (Harlan) were digested by trypsin (Amersham Biosciences) and collagenase type II (Worthington Biochemical). Isolated cells were re-suspended in the M-199 medium (Life Technologies) supplemented with 10% FBS, 19.4 mM glucose, 2 mM l-glutamine, 2 U/mL penicillin, 0.8 μg/mL vitamin B12, 10 mM HEPES, and 1 × MEM nonessential amino acids (Sigma-Aldrich). Two rounds of 60 min pre-plating were performed, during which cardiac fibroblasts attach to the dish bottom, thus enriching the nonadherent population for NRVMs. The nonadherent cells (NRVMs) were then seeded onto the different collagen matrices in 24-well plates (40,000 cells/cm²). For the survival assay, 3-day cultured NRVMs were subjected to M-199 media containing 50 μM hydrogen peroxide for 3 h, after which a Terminal deoxynucleotidyl transferase dUTP Nick-End Labeling (TUNEL) assay was performed and dead cells were counted in three random fields-of-view.

**Cell cultures**. Using an established protocol, bone marrow-derived macrophages were isolated from 8–12-week-old C57BL/6J mice[69]. Mice were euthanized by CO₂

inhalation and cervical dislocation, and tibia bones were collected and flushed with media to isolate the bone marrow. The bone marrow isolate was pipetted repeatedly to obtain a single-cell suspension, which was then passed through a cell strainer. Freshly isolated cells were cultured for 1 week in DMEM supplemented with 10% FBS, 20% L929-conditioned media, and penicillin–streptomycin, and then re-plated on the rHC hydrogels for 3 days. Cells were collected from the rHC hydrogels using 3 mM CaCl₂ Hank's buffer saline solution containing 250 units of collagenase I (Gibco). Macrophage polarization was assessed by flow cytometry (FACS Aria III; Becton Dickinson) using CD86 and CD206 (Biolegend) to identify M1 and M2 macrophages, respectively. For mononuclear cell isolation, bone marrow from tibia bones was collected as described above. Mononuclear cells were purified by density gradient centrifugation using Histopaque® (Sigma) according to the manufacturer's instructions. Cells were labeled with 0.5 μg/ml DAPI (Sigma) for 30 min at 37 °C.

**Macrophage adhesion assay**. Mononuclear cells were isolated by flushing the tibias and femurs of 6- to 12-week-old mice. Cells were purified by density gradient centrifugation using Histopaque® (Sigma) according to the manufacturers' instructions. Mononuclear cells were counted and labeled with DAPI. Cells were plated on the different biomaterials at a concentration of 50,000 cells/cm² for 24 h. Cells were fixed with 4% PFA for 5 min, and cells were counted. The data were expressed relative to the control (noncoated wells, i.e., TCPS) to minimize the effect of inter-donor cell variability.

**Macrophage migration assay**. Bone marrow mononuclear cells were cultured in DMEM supplemented with 10% FBS, 20% L929-conditioned medium, and Pen/Strep for 7 days to generate bone marrow-derived macrophages (BMDMs) and labeled with DAPI, as described above. BMDMs ($2 \times 10^5$) were then re-suspended in EBM lacking growth factors and serum, and loaded into the upper chamber of a Transwell plate (Life Technologies) coated with 100 μL of rHCI or rHCIII. The bottom chamber contained full macrophage media as described above. After 24 h, the inserts were removed, and the number of BMDMs that migrated through the biomaterial was quantified in a blinded fashion using a Zeiss Z1 fluorescence microscope. The data were expressed relative to the control (noncoated wells, i.e., TCPS) to minimize the effect of inter-donor cell variability.

**Macrophage polarization assay**. BMDMs were generated in DMEM supplemented with 10% FBS, 20% L929-conditioned medium, and Pen/Strep for 7 days, as described above. For macrophage activation experiments, cells were stimulated for 3 days with lipopolysaccharide (1 μg/ml; Sigma) plus IFN-γ (50 ng/ml; Sigma) for M1 activation, or with IL-4 (20 ng/ml; R&D systems) for M2 activation. The total RNA was extracted using Tri Reagent (Zymo Research), according to the manufacturer's protocol. First-strand cDNA was synthesized with Smartscribe Reverse Transcriptase (Takara Bio USA) and random hexamer primers (Fisher Scientific). Target gene mRNA levels were assessed by quantitative RT-PCR with LightCycler 480 SYBR Green I Mastermix (Roche) and a LightCycler 480 Real-Time PCR System (Roche). Sequences for primer pairs are listed in Supplementary Table 2. Relative changes in mRNA expression were determined by the ΔΔCt method, expressed as levels relative to 18S. The data were expressed relative to the control (noncoated wells, i.e., TCPS) to minimize the effect of inter-donor cell variability.

**Survival after H₂O₂ exposure**. Cells were cultured for 7 days in DMEM supplemented with 10% FBS, 20% L929-conditioned medium, and Pen/Strep. On day 7, the media was changed to DMEM containing 0.5 mM hydrogen peroxide, and cells were cultured for 3 h before being stained with 7-AAD for analysis by flow cytometry. The data were expressed relative to the control (noncoated wells, i.e., TCPS) to minimize the effect of inter-donor cell variability.

**Statistical analysis**. Statistical analysis was performed using Kaleida Graph 4.5®. All data are presented as the mean ± SEM. For the comparison of in vivo data between treatments, an ANOVA was used followed by Holm's correction for multiple comparisons. Scar size was analyzed using a multiple regression, including treatment group (rHCI, rHCIII, and PBS) and baseline LVEF. rHCI and rHCIII as compared with PBS were significant predictors of infarct size, in addition to baseline LVEF. The correlation coefficient for the model is 0.6 using STATA multiple linear regression. In vitro NRVM, macrophage and cardiac fibroblast data were analyzed either by a Student's t test or by a one-way ANOVA with a Holm's correction for multiple comparisons, as specified in the figure legends.

**Reporting summary**. Further information on research design is available in the Nature Research Reporting Summary linked to this article.

## Data availability

All data generated or analyzed in this study are included in the paper and the Supplementary materials. Source data are available in the Source Data file. The experimental data that support the findings of this study are available in figshare with the

identifier [https://doi.org/10.6084/m9.figshare.9873701]. For further request, please contact the corresponding authors.

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

## Acknowledgements

This work was supported by a Collaborative Research Grant from the Canadian Institutes of Health Research (CIHR) and the Natural Sciences and Engineering Research Council (NSERC) (CPG-158280 to E.J.S., E.I.A. and M.R.), a CIHR operating grant (MOP-77536 to M.R. and E.J.S.), an NSERC Discovery Grant (2015–06235 to E.I.A.) and a UOHI start-up grant (1255 to E.I.A.). S.M. was funded by a Frederick Banting and Charles Best Canada Graduate Scholarship; and K.G. by URS & UROP (University of Ottawa) scholarships.

## Author contributions

rHC development and SOPs: S.M., J.P. and K.G.; laboratory work: S.M., B.M. and K.H.; animal studies: S. M., B.M., V.S. and R.S.; ex vivo fluorescence imaging: G.C. and S.M.; flow-cytometry experiments and data analyses S.M., B.M. and D.S.; study design and trainee supervision: W.L., K.J.R., M.R., E.I.A. and E.J.S.; paper preparation: S.M., B.M., E.I.A. and E.J.S. with input from all authors.

## Competing interests

E.I.A., E.J.S. and M.R. are listed inventors in a patent application for the rHC materials presented in this study. Patent applicant: Ottawa Heart Institute Research Corporation, 40 Ruskin Street H2406 Ottawa, Ontario K1Y 4W7, Canada. Name of inventors: Emilio Alarcon, Erik Suuronen, Marc Ruel Application number: PCT/CA2018/050537 Status of the application: Published as WO/2018/201260. The specific aspect of manuscript covered in the patent application: Composition of matter for the hydrogel, a method for regenerating or repairing heart tissue, the method for preparing the composition of matter or the hydrogel, and the use of the hydrogel.
