## [Peer Review File · Nature Communications]

Reviewers' Comments:

Reviewer #1:

Remarks to the Author:

The manuscript by McLaughlin et al. examines the use of crosslinked recombinant human collagen injected into a mouse myocardial infarction model. Numerous injectable biomaterials have now been tested in MI models, including a few that have moved to clinical trials. The novelty of this study is the use of recombinant human collagen rather than animal derived collagen and specifically testing collagen I vs III. In general, the study is relatively thorough but there are some points that need to be addressed in order to fully evaluate the study.

Major comments:

The statement "Secondly, while biomaterials have prevented further loss of function when applied to the forming/maturing myocardial scar, they have not yet been shown to improve cardiac function" and the similar statement in the discussion are not correct assessments of the field. Some have shown to improve function.

While animal original indeed has some issues such as batch to batch variability, it is incorrect to say that being of animal original limits translation given the multitude of animal derived products that are approved by the FDA.

Pre- and post-treatment values for ESV and EDV should be reported. To fully evaluate the two materials impact on cardiac function and negative LV remodeling, these both need to be reported, not simply the change for only ESV.

It is unclear why the oxidative stress conditions were performed with macrophages rather than cardiomyocytes given that it is cardiomyocyte death that one is concerned with. Also, the mechanism for this protective effect is not discussed or elucidated.

In general, characterization regarding the chondroitin sulfate aspect of the materials is lacking/ignored. This addition could be leading to some if not a significant amount of the bioactivity. In vitro assays could be performed with and without this modification to determine its role.

How was successful material injection confirmed in the heart? There is no histology to show how the material looks in the heart.

The # of regions of slide analyzed should be stated. The methods used to quantify the IHC is likewise missing. In particular, it is unclear how borderzone was defined to provide an accurate assessment of cardiomyocyte area.

Given that the authors stress translatability it would be worthwhile to discuss the potential cost of a recombinant biomaterial therapy.

Minor:

Some of the references appear to be incorrectly #'ed.

Reviewer #2:

Remarks to the Author:

In this study authors report the development of an injectable hydrogel comprised primarily of recombinant human collagen. The authors describe the conception, synthesis, characterization and

application of recombinant human collagen type I (rHCI) and recombinant human collagen type III (rHCIII) matrices for “damage control” and cardiac repair after an acute myocardial infarction (MI). A mouse acute MI model was used for in vivo evaluation of therapeutic potential and matrix-cell interactions were characterized through in vitro assays examining connexin 43 coupling and macrophage polarization. The data presented in this manuscript show that injection of the rHCI matrix improved LV systolic function in a mouse acute MI model relative to injections of the rHCIII matrix, uncrosslinked recombinant human collagen or PBS.

This is an important area of research. Despite the myriad of novel drugs aimed at improving outcomes in patients suffering from HFrEF, the irreversible loss of functional cardiomyocytes and subsequent extracellular matrix remodeling following an acute MI leads to significant morbidity and mortality in this patient population. Prior animal model experiments in this field have shown the use of bovine-derived collagen and porcine-derived cardiac ECM in hydrogels to improve post-infarct LV function, but these face additional barriers to clinical translation due to the innate risks of pathogen transfer and xenogeneic immunogenicity associated with the use of animal collagens.

Over all this is a well-designed study, the use of recombinant human proteins can greatly accelerate the rate of clinical translation when compared with animal-derived proteins. Recombinant proteins also have the advantage of batch-to-batch consistency that facilitate development of an off-the-shelf product for widespread use. Following are my more specific comments;

1. Authors have injected human collagen to a mouse model, did they observe any xenogeneic immune response, authors should consider assessment of immune cell infiltration in the myocardial sections.
2. The biomaterial was injected one week after MI in a mouse model, the justification provided for this time point is that this is a model for patients who have delay in getting medical attention or have not adequately responded to conventional therapy. While this seems to be a valid point, authors should expand in the discussion that this one week post MI injection in mice translates to what age group in patients.
3. Authors need to further characterize mechanical properties of the hydrogel with respect to mechanical properties of native myocardium.
4. What was the final fate of biomaterial in the myocardium. Biodegradability of the hydrogel in vivo should be determined.
5. The authors state that healthy cardiac ECM is composed of 70% type I collagen and 12% type III collagen. The in vivo experiments showed that rHCI had a better therapeutic effect than rHCIII, yet in the in vitro assays presented in Figure 6, rHCIII showed higher M2 macrophage polarization than rHCI with associated higher resistance to H₂O₂-induced cell death. Authors should elaborate on this in the discussion.
6. The authors allude to the “counterintuitive” nature of injecting collagen. How is injected rHCI different than scar that forms naturally after an acute myocardial infarction? Were there any structural characterization performed comparing the differences in collagen structure between natural scar and injected rHCI in vivo? Does the heart replace the injected rHCI with autologous scar tissue over time?
7. Authors demonstrate that mobilization of bone marrow cells reduced in the heart treated to rHCI. However, it is reported that migration of bone marrow cells to the heart and other organs help in endogenous repair process. Authors should comment on this.

8. In vitro data suggests that connexin 43 expression increased in rat cardiomyocytes in the presence of biomaterial. This suggests that the hydrogel is able to improve electrical coupling and conduction velocity among remaining cardiomyocytes in the myocardium. Authors should have performed ECG after implantation of hydrogel in the heart. Also connexin 43 expression should be measured in vivo as well.

Reviewer comments are in *Italic*

REVIEWER #1

General comments

COMMENT #1-1: *The manuscript by McLaughlin et al. examines the use of crosslinked recombinant human collagen injected into a mouse myocardial infarction model. Numerous injectable biomaterials have now been tested in MI models, including a few that have moved to clinical trials. The novelty of this study is the use of recombinant human collagen rather than animal derived collagen and specifically testing collagen I vs III. In general, the study is relatively thorough but there are some points that need to be addressed in order to fully evaluate the study.*

Response: We thank the reviewer for his/her kind comments about our article. In the following pages, we provide a detailed explanation on how each comment has been addressed, including response and action (when applicable).

Action: No action required.

Specific comments

COMMENT #1-2: *The statement “Secondly, while biomaterials have prevented further loss of function when applied to the forming/maturing myocardial scar, they have not yet been shown to improve cardiac function” and the similar statement in the discussion are not correct assessments of the field. Some have shown to improve function.*

Response: We apologize for this misunderstanding. We meant to emphasize that most biomaterials are not effective at recovering cardiac function upon injection into a semi-mature myocardial scar. In reported cases where improvement has been seen, the treatment had occurred within hours post-MI. In this revised version, we reworded this paragraph and incorporated new references to other materials that improve cardiac function, albeit at much earlier time-points post-MI.

Action: The paragraph in question has been reworded, see pages 3 & 4, as follows:

“Secondly, while some biomaterials have prevented further loss of function when applied early to the forming myocardial scar^{17, 18, 19, 20, 21}, no collagen-based material has been shown to improve cardiac function in an established myocardial scar.”

COMMENT #1-3: *While animal original indeed has some issues such as batch to batch variability, it is incorrect to say that being of animal original limits translation given the multitude of animal derived products that are approved by the FDA.*

Response: We thank the reviewer for bringing up this point. We concur with some products of animal origin being both FDA approved and clinically used. However, concerns regarding the presence of endotoxins still

remain a challenge, making it more difficult to acquire regulatory approval for such products. In addition, with regards to myocardial repair there is no animal-derived injectable matrix available on the market. We have rewritten this sentence accordingly.

Action: The text on page 3 has been adjusted to reflect the limitations with the use of animal origin proteins in humans as follows:

“Despite the promise of biomaterials for cardiac therapy, there are still some limitations to consider. First, biomaterials tested so far have been of animal origin, which carry intrinsic batch-to-batch variability due to isolation protocols, and inherent immune risks (*i.e.* endotoxins), rendering their clinical translation challenging^{32,33}.”

COMMENT #1-4: *Pre- and post-treatment values for ESV and EDV should be reported. To fully evaluate the two materials impact on cardiac function and negative LV remodeling, these both need to be reported, not simply the change for only ESV.*

Response: As requested, the pre- and post-treatment ESV and EDV values have been added (Figures S3A & S3B), and the relative values for EDV have also been included in this revision (Figure 2D), as well as discussed. Specific details are provided below.

Action: As suggested by the reviewer, we have included the data for ESV (Figure S3A) and EDV (Figure S3B) measured at baseline and at 28 days post-treatment. The fold changes for EDV have also been added (Figure 2D). Accordingly, the following text has also been added to the results and discussion sections:

Results, page 7:

“For other parameters of cardiac function, the fractional area change (FAC) at 28 days relative to baseline was superior in rHCI-treated hearts compared to PBS and rHCIII treatment (Fig. 2B). The change in end-systolic volume (ESV) was reduced in rHCI-treated hearts compared to the other 2 groups (Fig. 2C), whereas no difference was observed for end-diastolic volume (EDV; Fig. 2D). ESV at 28 days was increased in the PBS group compared to rHCI matrix-treated mice (Fig. S3A), indicating worse remodeling and a worsening of cardiac function in the PBS-treated mice. For EDV at 28 days, no difference was observed between groups, but it was significantly increased for rHCIII-treated hearts at 28 days compared to its baseline (Fig. S3B).”

Discussion, page 18:

“The recovery of LVEF function was accompanied by reduced end-systolic volume, as well as improved fractional area change, stroke volume, and cardiac output, which are all indicators of overall improvement in cardiac function.”

COMMENT #1-5: *It is unclear why the oxidative stress conditions were performed with macrophages rather than cardiomyocytes given that it is cardiomyocyte death that one is concerned with. Also, the mechanism for this protective effect is not discussed or elucidated.*

Response: Since the intra-myocardial injection takes place at 7 days post-MI, the literature suggests that the number of viable cardiomyocytes in the scar tissue would be low. Furthermore, oxidative stress has been shown to regulate the inflammatory response in the post-MI heart. Thus, we initially had decided to explore the

oxidative effect on inflammatory cells, which are relatively abundant in the scar even 28 days post-injection. However, we agree with the reviewer that testing the effect of the rHC matrices on oxidative stress in cardiomyocytes would also be relevant. Therefore, we have added this new set of experiments and we discuss them accordingly.

Action: We performed additional *in vitro* experiments where neonatal rat ventricular myocytes (NRVMs) were cultured on the different substrates and exposed to 0.5 mM hydrogen peroxide (the same concentration that was used for macrophages). The data are plotted in Figure 7G, and the following text has been added to the results and discussion sections:

Results, page 17:

“The same H₂O₂ exposure experiments were performed with NRVMs; no protective effect from the collagen matrices was observed (Fig. 7G).”

Discussion, page 20:

“Oxidative stress post-MI is considered to be a major factor mediating the inflammatory cascade⁵⁸, and we therefore sought to examine the protective effect of the matrix against oxidative cell death in macrophages (BMDMs) and cardiomyocytes (NRVMs). Both rHCI and rHCIII matrices were able to reduce the number of death BMDMs in the presence of hydrogen peroxide. However, the NRVMs remain susceptible to H₂O₂-induced death when cultured on the rHC matrices. This suggests that our rHC materials may act as a protective niche limiting the loss of mononuclear cells in conditions of oxidative stress.”

COMMENT #1-6: *In general, characterization regarding the chondroitin sulfate aspect of the materials is lacking/ignored. This addition could be leading to some if not a significant amount of the bioactivity. In vitro assays could be performed with and without this modification to determine its role.*

Response: We thank the reviewer for raising this point. In the revision, we designed matrices without chondroitin sulfate and physically characterized them (see Figure S1). The absence of chondroitin sulfate had no effect on the denaturation temperature and porosity of the matrices. Furthermore, we performed new experiments for macrophage adhesion, and cardiomyocyte survival and Cx43 expression for matrices prepared without chondroitin sulfate (see Figure S7). These new data have been added as described below.

Action: A new Figure depicting the denaturation temperature and material porosity for collagen matrices prepared without chondroitin sulfate has been added, and is presented in the text as follows.

Results, page 6:

“The presence of chondroitin sulfate did not affect either the denaturation temperature or the porosity of the material (Fig. S1A & B).”

New data for cardiomyocyte and macrophage cultures on the matrices prepared without chondroitin sulfate are provided in Figure S7, and described in the results as follows:

Results, page 15:

“The presence of chondroitin sulfate did not affect connexin 43 expression in NRVMs for either of the

materials (Fig. S7A).”

Results, page 17:

“The adhesion of unstimulated M0 macrophages to the rHCI and rHCIII matrices was equivalent with (Fig. 7B) or without (Fig. S7B) the inclusion of chondroitin sulfate...”

COMMENT #1-7: *How was successful material injection confirmed in the heart? There is no histology to show how the material looks in the heart.*

Response: We thank the reviewer for raising this question. In delivering our matrices, we use an injection protocol with which our team has demonstrated and published experience, whereby the injection is monitored by echocardiography. Notably, we previously used PET imaging and *in vivo* fluorescence imaging to demonstrate the successful injection and retention properties of a similar rat-tail collagen hydrogel delivered to the post-MI mouse heart, using the same injection protocol that was used in the present study (Ahmadi *et al.*, *Biomaterials* 2015;49:18-26). However, we concur with the reviewer that a more illustrative example of the material injection would strengthen the present work. Thus, we have fluorescently labelled the rHC matrices for tracking their location post-injection using *ex vivo* total fluorescence organ imaging at 2h and 2 days post-injection. The protocol used and the new imaging data have been added, as explained below.

Action: Matrix labeling and imaging protocols have been added to the Methods section, and a new figure panel and paragraph describing the rHC matrix imaging is included in this revised manuscript as follows:

Results, page 6:

“To characterize the injection and retention properties of the collagen matrices, they were labeled with Alexa-Fluor®594-NHS dye prior to injection to the MI mouse heart. *Ex vivo* fluorescence imaging revealed that the injected rHCI and rHCIII matrices were localized and retained within the injection site for at least up to 2 days, which was confirmed by histological analysis of the infarct region (Fig. 1F).”

COMMENT #1-8: *The # of regions of slide analyzed should be stated. The methods used to quantify the IHC is likewise missing. In particular, it is unclear how borderzone was defined to provide an accurate assessment of cardiomyocyte area.*

Response: We better characterize our histological methodology, including how the different areas of infarction were defined. In this revised version, we expand on the histological methods description and better defined how the different tissues areas, including borderzone, were identified by using a schematic depiction (see below and Fig. S9).

Action: A new paragraph and figure further describing the histological methodology are included in this revised version of our article as follows:

Materials and Methods, pages 24 & 25:

“For all IHC stains, 4 sections per mouse were analyzed and for each section 2-4 images within the border zone, infarct and remote regions were used for analysis. The border zone was designated as the FOV directly adjacent

to either side of the infarct region which begins when 50% of the LV is fibrotic scar tissue (see Fig. S9 for a schematic depiction).”

COMMENT #1-9: *Given that the authors stress translatability it would be worthwhile to discuss the potential cost of a recombinant biomaterial therapy.*

Response: We thank the reviewer for this comment. An exact estimate of the costs of this therapy would be premature at this current stage, as expenses associated with GMP manufacturing are still undefined. Nevertheless, the average cost would be much lower than, for example, that of a ventricular assist device used in patients with advanced heart failure, which is around \$183,000 in Canada (see link for actual cost. This link will be active only for review purposes as the information is confidential). In this revised version, we further elaborate on the potential economic benefits of using our recombinant collagen-based therapy for cardiac tissue repair.

Action: A new paragraph describing the costs for treating patients with advanced heart failure in Canada and the monetary savings of our treatment has been now included in this revised version as denoted below:

Discussion, page 20:

“In terms of the potential cost savings from our new therapy, it should be considered that in Canada alone there are 600,000 patients living with advanced heart failure, whose healthcare costs amounts to more than \$2.8B every year⁵⁹. These patients require long and frequent hospital stays. Thus, a therapy that is able to reduce the progression to clinical heart failure would greatly improve both quality and quantity of life for many MI patients and potentially save millions in healthcare spending.”

Minor comments

COMMENT #1-10: *Some of the references appear to be incorrectly #’ed.*

Action: The references have been checked for spelling, numbering, and accuracy.

REVIEWER #2

General comments

Comment #2-1: In this study authors report the development of an injectable hydrogel comprised primarily of recombinant human collagen. The authors describe the conception, synthesis, characterization and application of recombinant human collagen type I (rHCI) and recombinant human collagen type III (rHCIII) matrices for “damage control” and cardiac repair after an acute myocardial infarction (MI). A mouse acute MI model was used for in vivo evaluation of therapeutic potential and matrix-cell interactions were characterized through in vitro assays examining connexin 43 coupling and macrophage polarization. The data presented in this manuscript show that injection of the rHCI matrix improved LV systolic function in a mouse acute MI model relative to injections of the rHCIII matrix, uncrosslinked recombinant human collagen or PBS.

This is an important area of research. Despite the myriad of novel drugs aimed at improving outcomes in patients suffering from HFrEF, the irreversible loss of functional cardiomyocytes and subsequent extracellular

matrix remodeling following an acute MI leads to significant morbidity and mortality in this patient population. Prior animal model experiments in this field have shown the use of bovine-derived collagen and porcine-derived cardiac ECM in hydrogels to improve post-infarct LV function, but these face additional barriers to clinical translation due to the innate risks of pathogen transfer and xenogeneic immunogenicity associated with the use of animal collagens. Over all this is a well-designed study, the use of recombinant human proteins can greatly accelerate the rate of clinical translation when compared with animal-derived proteins. Recombinant proteins also have the advantage of batch-to-batch consistency that facilitate development of an off-the-shelf product for widespread use.

Response: We thank the reviewer for his/her kind comments on our article. In the following pages, we present our responses and actions for each of the reviewer's comments.

Action: No action required.

Specific comments

COMMENT #2-2: *Authors have injected human collagen to a mouse model, did they observe any xenogeneic immune response, authors should consider assessment of immune cell infiltration in the myocardial sections.*

Response: We thank the reviewer for this point. As originally presented in Figure 5 using Cx3cr1-EGFP mice, we did not detect an increase in the recruitment of GFP⁺ bone marrow-derived monocytes within the infarcted myocardium at 2 days post-injection. However, in this revision, using flow cytometry, we further assessed the immune cell response in both the blood and hearts of wild-type mice at 2 days post-injection. These new data have been added to the results and discussion, as detailed below.

Action: New figures and a paragraph describing the flow cytometry data using wild type animals at 2-days post-treatment have been added as follows:

Results, pages 13 & 14:

“For further characterization of mobilized bone marrow monocytes, we evaluated peripheral blood mononuclear cells in wild-type mice 2 days after treatment. There was an increase in the overall number of circulating monocytes and a trend for increased numbers of Ly6C^{hi} monocytes in the rHCIII-treated mice compared to rHCI (Fig. 6A). No change in the other types of blood monocytes was seen between groups (Fig. S4). In the myocardium, the rHCIII matrix increased the number of Ly6C^{hi} monocytes 2 days after treatment, compared to PBS and rHCI (Fig. 6B), whereas there was no overall change in the numbers of leukocytes, macrophages, Ly6C^{lo} monocytes, neutrophils, T-cells or B-cells (Fig. S5). Finally, there was a higher retention of Ly6C^{lo} monocytes in the spleen for the rHCIII group compared to PBS (Fig. S6).”

Discussion, page 19:

“The analysis of monocyte populations 2 days post-treatment in wild-type mice showed an increase in the mobilization of monocytes into the blood and greater accumulation of Ly6C^{hi} cells in the myocardium for the rHCIII group. Also, an increased number of Ly6C^{lo} cells was seen in the spleen for the rHCIII group, suggesting some degree of inflammatory response to the rHCIII matrix, but not the rHCI matrix.”

COMMENT #2-3: *The biomaterial was injected one week after MI in a mouse model, the justification provided for this time point is that this is a model for patients who have delay in getting medical attention or have not adequately responded to conventional therapy. While this seems to be a valid point, authors should expand in the discussion that this one week post MI injection in mice translates to what age group in patients.*

Response: As requested, we added a statement indicating the relationship between the timeline of mouse MI compared to humans (*i.e.* a 7-day post-MI mouse heart corresponds approximately to 14 days in humans), as detailed below.

Action: In this revised version, we have added a new paragraph clarifying the reasoning for using mice at 7 days post-MI as follows (see pages 17 & 18):

“We have assessed the performance of our therapy at the end of the proliferative phase post-infarction, a time-point by which scar formation has been established (*i.e.* 7 days in mice, which is the equivalent of 14 days post-MI in humans)^{11, 37, 38, 39, 45}. This time-point represents a clinical opportunity to intervene to limit pathological ECM remodeling and promote infarct repair in patients who have not responded to surgical interventions and other therapeutics.”

COMMENT #2-4: *Authors need to further characterize mechanical properties of the hydrogel with respect to mechanical properties of native myocardium.*

Response: We thank the reviewer for this question. We would like to mention that measuring the mechanical properties of the myocardium using a rheometer, as we did for the collagen matrices, is not experimentally possible. This is because the minimum sample size for the rheometer must be about 4 cm in diameter and the sample must be flat, which is not feasible with murine cardiac tissue. Therefore, we have instead performed additional *in vivo* measurements using echocardiography to evaluate the mechanical properties (strain) of the cardiac muscle after treatment with the rHC matrices. These data provide a more representative picture of the actual effect that the rHC matrices have on the mechanical properties of the myocardium. We detected an improvement in the strain force values for hearts treated with the collagen matrices, compared to the controls. These new experimental data has been added to our article.

Action: A new experimental data set containing strain force values derived from echocardiography measurements for the different treatment groups had been added to the article (see Figure 2H). In addition, the following content has been added to the results and discussion sections, as follows:

Results, page 7

“*In vivo*, analysis of longitudinal endocardial strain through speckle tracking echocardiography⁴⁰ demonstrated a significant improvement in the strain reached by the mid anterior LV wall at end systole, which is marked by the aortic valve closure (AVC), two days after injection of rHCI (Fig. 2H). The mid anterior LV wall is the segment of the myocardium targeted for hydrogel injection as it contains the accessible infarct borderzone. The longitudinal endocardial strain becomes more negative during systole as the heart shortens in this direction due to the stress placed on the myocardium during contraction. In healthy animals, strain should peak at the AVC, which is an indicator of end systole and strain at this point is a measurement of myocardial contractility. Therefore, the strain analysis provides evidence that rHCI injection, but not rHCIII, improves contractility in the borderzone area of the LV wall where it was injected as compared to PBS treated animals. Neither rHC

matrix treatment affected the heart rate or any of the electrocardiographic parameters at 2 days post-injection (Table S1), with the exception of the PR interval for the rHCI matrix group. This indicates that the rHC matrices do not negatively interfere with electrical conductivity in the myocardium.”

Discussion, page 18:

“The mechanical properties of the myocardium, measured *ex vivo* (tensile strength), were comparably restored within days of receiving rHCI or rHCIII treatment, but only rHCI treatment improved the longitudinal endocardial strain measured *in vivo* (strain at the aortic valve closure), and both rHC treatments were accompanied by a decrease in the scar size”

COMMENT #2-5: *What was the final fate of biomaterial in the myocardium. Biodegradability of the hydrogel in vivo should be determined.*

Response: We have used a fluorescent labeling technique for monitoring the distribution of the rHC matrices post-injection *in vivo* for up to 2 days. Please see our response to Reviewer#1; **COMMENT #1-7** for further details.

Action: Our action has been explained above; see response to comment #1-7.

COMMENT #2-6: *The authors state that healthy cardiac ECM is composed of 70% type I collagen and 12% type III collagen. The in vivo experiments showed that rHCI had a better therapeutic effect than rHCIII, yet in the in vitro assays presented in Figure 6, rHCIII showed higher M2 macrophage polarization than rHCI with associated higher resistance to H2O2-induced cell death. Authors should elaborate on this in the discussion.*

Response: We acknowledge some degree of confusion on this point. We would like to mention that there was no significant difference in the resistance to hydrogen peroxide for macrophages incubated on rHCI or rHCIII. As for the greater polarization of M2 macrophages observed with rHCIII, this could not be directly extrapolated to the *in vivo* setting. We have revised the discussion and data presentation for this part of the study in our revised version.

Action: Text in the discussion section has been amended to emphasize that there was no difference in the protective effect from the collagen matrices against the hydrogen peroxide oxidation, see page 20:

“Oxidative stress post-MI is considered to be a major factor mediating the inflammatory cascade⁵⁸, and we therefore sought to examine the protective effect of the matrix against oxidative cell death in macrophages (BMDMs) and cardiomyocytes (NRVMs). Both rHCI and rHCIII matrices were able to reduce the number of death BMDMs in the presence of hydrogen peroxide. However, the NRVMs remain susceptible to H₂O₂-induced death when cultured on the rHC matrices. This suggests that our rHC materials may act as a protective niche limiting the loss of mononuclear cells in conditions of oxidative stress.”

In the discussion, we also include the following to address the difference in M2 macrophage results between the *in vitro* and *in vivo* studies (pages 19 & 20):

“Both rHC matrices promoted greater M2 polarization and expression of M2 markers (MMP1 and Arg1) than TCPS, but more so with the rHCIII matrix than the rHCI matrix. This is in contrast to our *in vivo* observations, in which only the rHCI matrix, but not rHCIII, led to greater numbers of M2 macrophages in the infarcted heart

long-term compared to PBS control treatment. This may be related to differences between the *in vitro* and *in vivo* conditions. For example, it has been shown that M2 polarization is stimulated to a greater extent in 3D vs. 2D environments⁵⁷. Thus, it is plausible that, although cells are seeded on the surface of the materials *in vitro* (*i.e.* in 2D), the larger pore size of the rHCIII material may result in more cells being exposed to a 3D-like environment leading to greater M2 macrophage polarization. In the *in vivo* setting, the injected material is subjected to enzymatic degradation, which was shown to be greater for the rHCI material; thus, we speculate that this leads to greater cell invasion and an increased number of M2 macrophages over time in the rHCI treatment group.”

COMMENT #2-7: *The authors allude to the “counterintuitive” nature of injecting collagen. How is injected rHCI different than scar that forms naturally after an acute myocardial infarction? Were there any structural characterization performed comparing the differences in collagen structure between natural scar and injected rHCI in vivo? Does the heart replace the injected rHCI with autologous scar tissue over time?*

Response: We thank the reviewer for this excellent question. Post-MI, alterations of ECM proteins, including collagens, are initiated during the inflammatory phase (Bayomy *et al.*, *Life Sci* 2012;91:823-7; Barallobre-Barreiro *Circulation* 2012;125:789-802; Dobaczewski *et al.*, *J Mol Cell Cardiol* 2010;48:504-11). These changes disrupt the cell-ECM interactions required for cell signaling, function and survival. Our injected collagen materials have not been subjected/modified by the post-MI environment and upon injection can restore some of the cell-ECM interactions (Ahmadi *et al.*, *Biomaterials* 2014;35:4749-58). We looked for an antibody capable of distinguishing between autologous type I & III collagens (mice) and our injected collagen matrices (human). However, due to the high homology of the collagen structures between species (see <https://doi.org/10.1093/molbev/msq221>), finding an antibody with suitable specificity, as per indicated by the manufacturers, capable to provide reliable experimental data was not possible. While long-term tracking and remodeling of the injected matrix is an interesting and important consideration, we currently do not have suitable methods to address this question.

Action: No action could be taken.

COMMENT #2-8: *Authors demonstrate that mobilization of bone marrow cells reduced in the heart treated to rHCI. However, it is reported that migration of bone marrow cells to the heart and other organs help in endogenous repair process. Authors should comment on this.*

Response: We have further explored the cells mobilized from the bone marrow and the presence of inflammatory cells in the infarcted myocardium. We have focused our studies on the inflammatory cells of the bone marrow (leukocytes), given that we had observed increased wound healing macrophages in our rHCI treated mouse hearts, and reduced inflammation in our previous work (Blackburn *et al.*, *Biomaterials* 2015;39:182-192). While other components of the bone marrow may also be playing a role in the therapeutic effect of our rHC matrices (such as bone marrow c-kit⁺ cells), this may constitute an area for future investigation. The new data has been added in Figure 6; see response above to **COMMENT #2-2**.

Action: See action in response to comment #2-2.

COMMENT #2-9: *In vitro* data suggests that connexin 43 expression increased in rat cardiomyocytes in the presence of biomaterial. This suggests that the hydrogel is able to improve electrical coupling and conduction velocity among remaining cardiomyocytes in the myocardium. Authors should have performed ECG after implantation of hydrogel in the heart. Also connexin 43 expression should be measured in vivo as well.

Response: Once again, we thank the reviewer for their comment. As requested, we have included measurements for connexin 43 levels in cardiac tissue sections (see Figure 4C). In addition, we have provided ECG data for hearts at 2 days post-treatment, and overall, there was no effect of matrix treatment on the various tested parameters (see new Table S1).

Action: A new experimental data set for connexin 43 (Cx43) levels in the remote region has been added in this revised version and new sentences incorporated in the results and discussion sections as follows:

Results, page 10:

“Notably, the level of connexin 43 expression in the remote zone was greater for the rHCI matrix group compared to PBS (Fig. 4C).”

Discussion, page 19:

“This may be an explanation for the increased expression of cardiac troponin I in the border zone and Cx43 in the remote region of rHCI-treated hearts compared to either PBS or rHCIII.”

In addition, the newly measured ECG data has been included in the results section as follows (Results, page 7): “Neither rHC matrix treatment affected the heart rate or any of the electrocardiographic parameters at 2 days post-injection (Table S1), with the exception of the PR interval for the rHCI matrix group. This indicates that the rHC matrices do not negatively interfere with electrical conductivity in the myocardium.”

*Once again, we would like to thank the reviewers for their insightful comments, which helped to improve the quality, clarity, and impact of our manuscript.
Sincerely,*

Reviewers' Comments:

Reviewer #1:

Remarks to the Author:

The authors have been mostly responsive to the previous critiques but a few issues still remain:

The authors are still mistaken regarding the field of injectable biomaterials. Improvements have been seen in a semi-mature scar, not just injections acutely post-MI. The sentence starting with "Secondly, while some biomaterials..." should be completely removed. Same with the corresponding sentence in the discussion ("but such an improvement has not been previously reported for any biomaterial treatment administered to a more advanced scar") In addition, the current study is injecting only 1 week post-MI which is also not an established scar. This likewise needs to be adjusted throughout the text.

Also, it is fine to say the limitations of animal derived materials but "rendering clinical translation challenging" from that sentence should be removed. This misconception in the field should not be propagated. There are currently no injectable products approved for this indication, either synthetic, animal derived, or human derived. The FDA is very accustomed to animal derived materials and any new material will have regulatory challenges.

The authors have not added biodegradability assessment per Reviewer 2's comment, which is important given this is a new material in the heart. The only assessment is out to day 2. Showing degradation time would be very valuable, and is an easy experiment.

The authors have added fluorescent imaging of the injected collagens showing retention which is good, but it would still be helpful to have an H&E image of the 2 hr post-injection time point to better visualize what the injected materials look like in the infarct. As currently presented in 1F, it is difficult to visualize the material and in fact it does not look like any is in the infarct, only a few dots on the epicardium.

Changes in EDV are the hallmark of negative LV remodeling. However, there were no changes with either material injection compared to saline. The discussion therefore needs to be modified. For example, the authors cannot say that enlargement or negative LV remodeling of the heart was fully prevented. It is still interesting that there are changes in ESV, but there needs to be a discussion of why that might be affected but not EDV.

It is still not clear why macrophage cell death is being looked at. The paper (ref 58) that is cited does not mention any issues with macrophages death.

Reviewer #2:

Remarks to the Author:

No further comments

Reviewer comments are in *Italic*

REVIEWER #1

General comments

COMMENT #1-1: The authors have been mostly responsive to the previous critiques but a few issues still remain.

Response: We thank the reviewer for his/her kind comments about our article. In the following pages, we provide a detailed explanation on how each comment has been addressed, including response and action (when applicable).

Action: No action required.

Specific comments

COMMENT #1-2: The authors are still mistaken regarding the field of injectable biomaterials. Improvements have been seen in a semi-mature scar, not just injections acutely post-MI. The sentence starting with “Secondly, while some biomaterials...” should be completely removed. Same with the corresponding sentence in the discussion (“, but such an improvement has not been previously reported for any biomaterial treatment administered to a more advanced scar”) In addition, the current study is injecting only 1 week post-MI which is also not an established scar. This likewise needs to be adjusted throughout the text.

Response: We apologize for the wording of our article, which we have tried to clarify. In our search of the literature, we found studies of injectable materials delivered during the late proliferative phase of MI that improved function in the heart compared to controls, but none that resulted in greater function relative to their own baseline function at the time of delivery. We meant to highlight that injectable therapeutic biomaterials have been shown to prevent further functional loss, but have not led to increased cardiac function when used in infarcted hearts in the late proliferative phase post-MI. We have reworded the claim to more simply state that such materials are needed.

We concur with the reviewer that the wording used to describe the stage of the scar might be misinterpreted. At the time that the treatment was delivered (7 days post-MI), the scar has been formed, and so we referred to this scar as being established. This would be different than the mature scar that would appear up to weeks later. However, in order to improve our accuracy, where appropriate, we have removed descriptive terms for the scar and have replaced them with the stage of infarction (*e.g.* we use the term late proliferative phase rather than established scar to describe the time-point at which the treatment was delivered). In this revised version, we have reworded the relevant sections as indicated below.

Action: The paragraph in question has been reworded, see pages 3 & 4, as follows:

“While biomaterials have prevented further loss of function when applied early post-MI^{17, 18, 19, 20, 21}, an injectable material that can increase cardiac function when delivered during the late proliferative phase post-MI is also needed.”

In addition, all other relevant sections in the discussion have been reworded (see pages 18 & 19), including:

“Many have been shown to reduce adverse remodeling and limit the loss of function post-MI; but a material is needed that can increase cardiac function when used as a stand-alone therapy for treating myocardial scar at the late proliferative stage^{41,42,43}.”

“We have assessed the performance of our therapy at the end of the proliferative phase post-infarction (*i.e.* 7 days in mice, which is the equivalent of 14 days post-MI in humans)^{11, 37, 38, 39, 45}. This time-point represents a clinical opportunity to intervene to limit pathological ECM remodeling and promote infarct repair in patients who have not responded to surgical interventions and other therapeutics.”

“The improvement in LVEF for the rHCI group was similar to that observed in MI mice that received rat-tail collagen matrix treatment delivered acutely at 3h post-MI²⁶.”

COMMENT #1-3: *Also, it is fine to say the limitations of animal derived materials but “rendering clinical translation challenging” from that sentence should be removed. This misconception in the field should not be propagated. There are currently no injectable products approved for this indication, either synthetic, animal derived, or human derived. The FDA is very accustomed to animal derived materials and any new material will have regulatory challenges.*

Response: We thank the reviewer for bringing up this point. Our intention was to stress that the pre-regulatory process for clinical trials are much more challenging when using animal derived products, particularly in Canada, where cardiovascular implants together with meeting ISO 10993 standards and being manufactured under GMP standards for clinical testing must meet the highest level of safety (Class IV). The intrinsic issues from batch-to-batch variability for animal origin derived products (such as proteins) contribute to this challenge. However, we concur with the reviewer that the portion of the text stating “rendering their clinical translation challenging” might have sounded too inflammatory. Thus, we have removed this part as requested by the reviewer.

Action: The text on page 3 has been adjusted as requested by the reviewer, as follows:

“Despite the promise of biomaterials for cardiac therapy, there are still some limitations to consider. First, biomaterials tested so far have been of animal origin, which carry intrinsic batch-to-batch variability due to isolation protocols, and inherent immune risks (*i.e.* endotoxins)^{32, 33}.”

COMMENT #1-4: *The authors have not added biodegradability assessment per Reviewer 2’s comment, which is important given this is a new material in the heart. The only assessment is out to day 2. Showing degradation time would be very valuable, and is an easy experiment.*

Response: Our previously reported data presented in Fig. 1F was collected for up to 2 days, and showed that rHCIII hydrogels degraded to a greater extent after injection into the post-MI heart compared to rHCI. In fact, very little of the rHCIII hydrogel remained 2 days after its delivery. In this revised version, we have incorporated new IVIS data to further characterize the degradation of rHCI at 7 days post-MI. The results show that at 7 days post-injection, the rHCI has reached levels close to those observed for rHCIII at day 2. This data suggests that that rHCI degrades considerably slower than rHCIII, and is $\approx 80\%$ gone by 1 week after delivery.

Action: As suggested by the reviewer, we have included new IVIS data for the degradation of rHCI 7 days post-MI. Accordingly, the following text has been modified in the results section, page 6:

“*Ex vivo* fluorescence imaging revealed that the injected rHCI and rHCIII matrices were localized and retained within the injection site for at least up to 2 and 7 days, respectively. This was also confirmed by histological analysis of the infarct region using the labelled matrix fluorescence emission (Fig. 1F). Based on this labeling technique, the injected rHC matrices appear to localize primarily within tissue on the epicardial side, which was also observed in fluorescent imaging and H&E staining of tissue sections at 2h post-delivery (Fig. S2).”

Also, the following paragraph has been added to the experimental section, pages 24 & 25:

“For H&E staining, tissue sections were fixed in 10% formalin. Sections were then stained first in hematoxylin gill’s solution No. 2 for 7 min, followed by acid alcohol differentiation, and then 0.5% eosin for 7 min, followed by dehydration (all solutions from Sigma). Images were taken with an Olympus BX50 microscope.”

COMMENT #1-5: *The authors have added fluorescent imaging of the injected collagens showing retention which is good, but it would still be helpful to have an H&E image of the 2 hr post-injection time point to better visualize what the injected materials look like in the infarct. As currently presented in 1F, it is difficult to visualize the material and in fact it does not look like any is in the infarct, only a few dots on the epicardium.*

Response: We thank the reviewer for this comment. In this revised version, we have added H&E images, see Fig. S2.

Action: As requested, H&E images of the treated hearts at 2h post-injection have been added. In the response to COMMENT #1-4 above, the new text that has been modified to incorporate such data is further explained.

COMMENT #1-6: *Changes in EDV are the hallmark of negative LV remodeling. However, there were no changes with either material injection compared to saline. The discussion therefore needs to be modified. For example, the authors cannot say that enlargement or negative LV remodeling of the heart was fully prevented. It is still interesting that there are changes in ESV, but there needs to be a discussion of why that might be affected but not EDV.*

Response: The reviewer correctly indicates that a change in EDV has been a hallmark of negative cardiac remodeling. However, this prognostic indicator is typically used in assessing adverse remodeling months after the MI has occurred (4-6 months in humans), and our furthest time-point evaluated is at 5 weeks post-MI (4 weeks post-treatment). Furthermore, this clinical end-point alone, established in 1986, is no longer considered precise given the new information being gathered about ventricular remodeling (*Cokkinos and Belogiannas, Eur Cardiol Rev 2016;11:29-35; Cohn et al. J Am Coll Cardiol 2000;35:569-82*). Notably, ESV has been correlated with the rate of functional improvement after coronary artery bypass grafting (*Mandegar et al. J Thorac Cardiovasc Surg 2008;136:930-6*), ESV measured at 4-8 weeks after MI (closer to our experimental time-points) was a greater predictor for survival than EDV (*White et al. Circulation 1987;76:44-51*), and ESV was an early predictor for progressive ventricular dysfunction and remodeling post-MI (*Gaudron et al. Circulation 1993;87:755-63*). Given this evidence, we do not feel that it is necessary to include additional discussion about the lack of a difference in EDV values seen in our study.

Action: No action required.

COMMENT #1-7: *It is still not clear why macrophage cell death is being looked at. The paper (ref 58) that is cited does not mention any issues with macrophages death.*

Response: We thank the reviewer for raising this question. As shown in Fig. 5A, animals treated with rHCI showed an increased number of CD206+ cells in the myocardium, which led us to hypothesize that this treatment was able to protect and/or recruit these pro-wound healing macrophages. Thus, the *in vitro* experiments shown in Fig. 7F aimed to test the protective effect of our matrices on macrophages when incubated with hydrogen peroxide, which mimics the cell death-inducing oxidative stress conditions present in the infarcted myocardium. In relation to this, reference 58 describes the role of oxidative stress in cardiac remodeling, the condition we expose our cells to in order to mimic the infarct environment. However, further review of the paragraph in question has led us to conclude the wording could be improved.

Action: The paragraph discussing the effect of hydrogen peroxide on cell survival has been modified as follows:

Discussion, page 20:

“Oxidative stress post-MI is considered to be a major factor in cardiac remodeling.⁵⁸ We therefore sought to examine the protective effect of the matrix against oxidative stress-induced death of macrophages (BMDMs), which were increased in number within the myocardium of rHCI-treated hearts, and cardiomyocytes (NRVMs).”

Once again, we would like to thank the reviewer for his/her insightful comments, which helped to improve the quality, clarity, and impact of our manuscript.

Sincerely,

Emilio I. Alarcon, Ph.D., MSc, BSc
Principal Investigator/Assistant

Erik J. Suuronen, Ph.D.
Principal Investigator/Associate Professor

Reviewers' Comments:

Reviewer #1:

Remarks to the Author:

The images in Figure S2 show only extremely small injection areas that are not in the infarct. It does not appear to be a successful injection. A successful injection would see a significant amount of material in the infarct/borderzone at 2 hours post-injection. There are numerous studies in the literature with all types of materials, including collagen, that show this. This is thus concerning related to the claims of the paper.

Also the reviewer's correlation of the rat infarct timing directly with humans is incorrect. Everything in the rat is accelerated. There should still be some discussion addressing the original comment related to changes in ESV vs EDV.

All other comments have been satisfactorily addressed.

Reviewer#1 comments in *Italic*

Comment #1-1: *The images in Figure S2 show only extremely small injection areas that are not in the infarct. It does not appear to be a successful injection. A successful injection would see a significant amount of material in the infarct/borderzone at 2 hours post-injection. There are numerous studies in the literature with all types of materials, including collagen, that show this. This is thus concerning related to the claims of the paper.*

Response #1-1: We thank the reviewer for bringing to our attention that the injection and retention properties of our materials needed further elaboration. In our response below, we explain how the presence of the injected material was found primarily within the epicardium (rather than penetrating throughout the myocardium), and that it spread across the border zone and infarct areas after injection. We have provided additional images and a video to further highlight and confirm these injection/retention properties.

The histological sections shown in Fig. S3 (formerly S2) are 2D images of the myocardium post-injection and show the epicardial location of the material at 2 hours post-injection. While penetration into the deeper myocardium is not observed, the rHC matrix can be seen within the epicardial tissue. By zooming in on any part of the image in these areas, the rHC material can be seen in the epicardium from the border zone on one side of the infarct area all the way across to the border zone on the other side. To further demonstrate this, new fluorescence images of sections taken at different depths of the rHC-treated hearts are provided in Fig. S3C & D. These images were taken of sections that started at around 1.44 mm from the Apex and increased in 0.48 mm increments towards the base of the heart; they show the distribution of the injected material in the epicardium of the border zone and infarct across all these regions at 2 hours post-injection. The spread of the rHC matrices is best shown by the IVIS *ex-vivo* imaging (Fig. 1F), which depicts the successful injection, retention and distribution of the injected materials. Specifically, the *in vivo* imaging reveals retention of the injected rHC material at 2 hours, which is then shown to be spread further across the myocardium at 2 days. The IVIS images also demonstrate the degradation and loss of the injected material over time (up to 7 days). Taken together, we think that these histological and *in vivo* imaging data convincingly shows successful injection, localization, and retention of our material. To help further clarify the injection of the material, we have added a video (Video S1), as well as some captured images from the video (Fig. S1), depicting how the echo-guided injections are carried out.

Furthermore, although not explicitly stated by the reviewer, perhaps the concern regarding the injection relates to its lack of penetration into the myocardium since the histological analysis presented in Fig. S3 (formerly S2) shows that the injected material is primarily located within the epicardium. However, we can find no evidence in the literature to conclude that a material needs to be retained deeper within the myocardium in order to exert its therapeutic effects. To the contrary, previous work has demonstrated that epicardial treatments can repair/regenerate the infarcted myocardium. For example, pre-formed cardiac patches applied to the epicardial surface of the infarcted heart have been

therapeutically effective despite their lack of penetration into the tissue (*e.g.*, Shah *et al.* ACS Appl Mater Interfaces. 2019 Jun 28 [e-pub ahead of print; ACS Appl Mater Interfaces. 2019 Jun 28. doi: 10.1021/acsami.9b06453]; Liang *et al.* Adv Mater 2018;30:e1704235; D'Amore *et al.* Biomaterials 2016;107:1-14; Serpooshan *et al.* Biomaterials 2013;34:9048-55). Furthermore, Wei *et al.*, demonstrated that activating epicardial cells was an effective way to reverse myocardial death and remodelling post-MI (Nature 2015;525:479-485). Our rHC matrices are injectable and are able to exert their positive effects on MI wound healing and cardiac function through retention within the epicardium.

Actions in response to comments #1-1:

The following actions have been taken to elaborate on the injection, retention and distribution properties of the rHC materials:

A new supplementary figure (Fig. S1) and video (Video S1) have been added to this revised version of our article. The purpose is to better depict how the material is delivered to the myocardium under echo-guidance. The following sentences have been added/revised in the manuscript to explain this:

- On page 4, lines 97-103 have been reworded as follows: “The pre-clinical performance of our injectable rHCI and rHCIII materials was tested in a well-established mouse model of MI³⁵. MI mouse hearts received intramyocardial injections of rHCI or rHCIII administered at 1-week post-MI (Fig. S1 and Video S1), which is a model for patients who have a delay in getting medical attention (*e.g.* remote location, or ignore early signs after heart attack) or who have not adequately responded to conventional treatment, and are at greater risk of ECM remodeling and developing heart failure^{36,37}.”
- On page 23, lines 607-610 have been reworded as follows: “At 1 week post-MI (baseline), mice were randomly assigned to receive treatment of PBS (control), rHCI or rHCIII matrices, delivered in 5 equivolumetric intramyocardial injections (10µl each site, 50µl total) through a 27G needle using an ultrasound-guided closed-chest procedure (see Fig. S1 and Video S1)^{26,27}...”

New supplementary histological images (Fig. S3C & D) have been added. These images correspond to sections taken at different depths across the myocardium from the apex towards the base and show how the injected matrices spread across the border zone and infarct areas.

- On page 6, lines 151-157, have been reworded as follows: “This was also confirmed by histological analysis of the infarct region using the labelled matrix fluorescence emission (Fig. 1F). Based on these labeling techniques, the injected rHC matrices appear to localize primarily within the epicardial tissue spreading across the infarct and border zone areas. This was also observed in fluorescent imaging and H&E staining of tissue sections at an earlier 2h post-delivery time-point (Fig. S3A & B). Sections at different depths from the apex to the base of the

rHC-treated hearts further confirm the presence of the rHC material within the epicardium and its spread over the infarct and border zone areas at 2h post-injection (Fig. S3C & D).”

Comment #1-2: *Also the reviewers correlation of the rat infarct timing directly with humans is incorrect. Everything in the rat is accelerated.*

Response #1-2: While we cannot find any statement in which we directly correlate the healing rate between humans and mice (mice were used, not rats), we did indeed acknowledge the accelerated healing time-line in our animal model compared to humans, in the original submission, as follows:

Page 4: “This time-point constitutes the proliferative phase of infarction during which scar formation is initiated (mice: 2-7d; humans: 4-14d) and is a prime opportunity to intervene...”

Page 18: “We have assessed the performance of our therapy at the end of the proliferative phase post-infarction (*i.e.* 7 days in mice, which is the equivalent of 14 days post-MI in humans)...”

Action in response to comment #1-2: Since we already indicated that the infarct healing process is accelerated in mice compared to humans in the previous submission, we feel no further action is required at this stage.

Comment #1-3: *There should still be some discussion addressing the original comment related to changes in ESV vs EDV.*

Response #1-3: We have added a discussion section in this revised version to address the topic.

Action in response to comment #1-3: The following content has been added to the discussion section (Page 19, lines 462-467): “In contrast, end-diastolic volume (EDV) was not improved in hearts that received rHC treatment. However, this prognostic indicator is typically used in assessing adverse remodeling months after MI has occurred (4-6 months in humans; equivalent to ~2-3 months in mice), and our furthest time-point evaluated is at 5 weeks post-MI (4 weeks post-treatment). Furthermore, while the use of EDV alone as a clinical end-point has been questioned, ESV has been shown to be an early predictor for recovery of the MI heart^{51, 52, 53, 54, 55}”

Together with thanking the reviewer for his/her insightful comments, we feel that our data strongly supports the successful injection and epicardial retention of our material, which is the main concern of Reviewer #1. As such, we remain confident that this manuscript will be well-received and of great interest to the readers of Nature Communications. Should you require any additional information, please do not hesitate to contact us.

Reviewers' Comments:

Reviewer #2:

Remarks to the Author:

Authors have successfully addressed the reviewer's concerns, I have no further comments.

Reviewer #3:

Remarks to the Author:

The authors have presented a body of work in which their injectable hydrogels have durability to be injected onto the epicardium of an infarcted heart. They show improvements in cardiac function, decreased reserves remodeling, promotion/recruitment of wound healing macrophages, and decreased inflammatory cells in the hearts treated with an injectable hydrogel compared to vehicle control.

They have adequately responded to the previous reviewers' questions and the additional data, figures and analysis add to their verbal rebuttal.

There are a few (minor) issues that have not been addressed previously. Please explain the following issues:

- 1) Your neonatal rat VM data in figure 7A is weak. The Cx43 may be expressed but the distribution intracellularly and between cells is not meaningful. If you want to demonstrate cell connectivity, choose alpha-integrin instead
- 2) Why don't you show TCPS (control) data for quantified Cx43? What was the purpose of pacing the cells? Explain this in your methods or in the results
- 3) In all of your figures, please explain (each figure legend) what your "n" refers to: number of samples, number of animals, how many samples per animal? There are a lot of data elements here, so it is not implicit what each of your "n" refer to.

REVIEWER#2: Comments in *italic*

Comment #2-1: *Authors have successfully addressed the reviewer's concerns, I have no further comments.*

Response #2-1: We would like to thank the reviewer for his/her time and dedication to read our article and for providing us with valuable feedback during the review process.

REVIEWER#3 comments in *Italic*

Comment #3-1: *The authors have presented a body of work in which their injectable hydrogels have durability to be injected onto the epicardium of an infarcted heart. They show improvements in cardiac function, decreased reserves remodeling, promotion/recruitment of wound healing macrophages, and decreased inflammatory cells in the hearts treated with an injectable hydrogel compared to vehicle control. They have adequately responded to the previous reviewers' questions and the additional data, figures and analysis add to their verbal rebuttal.*

Response #3-1: We thank the reviewer for considering our responses and body of work suitable for publication in Nature Comm. In the following, we have responded to the reviewer additional comments.

Comment #3-2: *Your neonatal rat VM data in figure 7A is weak. The Cx43 may be expressed but the distribution intracellularly and between cells is not meaningful. If you want to demonstrate cell connectivity, choose alpha-integrin instead.*

Response #3-2: We have added new experimental data in which we analyzed integrin $\alpha 5$ expression in neonatal rat VM (see Supplementary Figure 10). Accordingly, new content has been added to the results, discussion and methods sections, as outlined below:

Actions in response to comments #3-2: The following sections have been added into this revised version of the article:

Results. On page 16 we have added the sentence “Similar results were observed when assessing integrin $\alpha 5$, with increased expression observed in NRVMs cultured on rHCI under electrical stimulation (Supplementary Fig. 10).”

Discussion. On page 20 we write “In addition, higher levels of integrin $\alpha 5$ were observed for cells cultured under electrical stimulation on the rHCI matrix.”

Methods. On page 27 we have updated the following sentence: “Cells were fixed with 4.0% PFA and stained with mouse anti-alpha sarcomeric actinin antibody (α -SA; 1:400, Sigma) and either rabbit anti-connexin 43 antibody (Cx43; 1:200, Sigma) or rabbit anti-mouse alpha 5 integrin ($\alpha 5$ integrin, 1:250, Abcam)^{70, 71, 72}”

Comment #3-3: *Why don't you show TCPS (control) data for quantified Cx43? What was the purpose of pacing the cells? Explain this in your methods or in the results.*

Response #3-3: In the bone marrow-derived macrophage (BMDM) culture studies, the data was normalized for each experimental set to its own control group (TCPS) in order to minimize the effect of inter-donor variability in the primary cells that were isolated from different mice. To maintain consistency in the presentation of the data in the figure, the NRVM results are presented in the same way as the BMDM results. As for the purpose of pacing the cells, we aimed to mimic the endogenous electrical stimulation that cells are exposed to within the heart.

Actions in response to comments #3-3: The following sections have been added to this revised version of the article:

Methods. Page 27, “...electrical stimulation. To more closely mimic the endogenous environment within the cardiac muscle, cells were electrically stimulated for 24h...”

Methods. Page 28/29, the following statement was added to the relevant methods sections “Data was expressed relative to the control (non-coated wells, i.e. TCPS) to minimize the effect of inter-donor cell variability”.

Comment #3-4: *In all of your figures, please explain (each figure legend) what your “n” refers to: number of samples, number of animals, how many samples per animal? There are a lot of data elements here, so it is not implicit what each of your “n” refer to.*

Response #3-4: In this revision, we have clarified the meaning of “n” in each Figure of our article, as requested.

Actions in response to comments #3-4: The meaning of “n” has been clarified in each figure of our article.

We thank the reviewer for his/her insightful comments.

Sincerely,